

# Universal front propagation in the quantum Ising chain with domain-wall initial states

V. Eisler[1,2], F. Maislinger[1] and H. G. Evertz[1,3]

**1** Institut für Theoretische Physik, Technische Universität Graz, Petersgasse 16, A-8010 Graz, Austria
**2** MTA-ELTE Theoretical Physics Research Group, Eötvös Loránd University, Pázmány Péter sétány 1/a, H-1117 Budapest, Hungary
**3** Kavli Institute for Theoretical Physics, University of California, Santa Barbara, CA 93106, USA

## Abstract

We study the melting of domain walls in the ferromagnetic phase of the transverse Ising chain, created by flipping the order-parameter spins along one-half of the chain. If the initial state is excited by a local operator in terms of Jordan-Wigner fermions, the resulting longitudinal magnetization profiles have a universal character. Namely, after proper rescalings, the profiles in the noncritical Ising chain become identical to those obtained for a critical free-fermion chain starting from a step-like initial state. The relation holds exactly in the entire ferromagnetic phase of the Ising chain and can even be extended to the zero-field XY model by a duality argument. In contrast, for domain-wall excitations that are highly non-local in the fermionic variables, the universality of the magnetization profiles is lost. Nevertheless, for both cases we observe that the entanglement entropy asymptotically saturates at the ground-state value, suggesting a simple form of the steady state.


# 1 Introduction

The nonequilibrium dynamics of isolated many-body systems is at the forefront of developments within quantum statistical physics research, see Refs. [1–3] for recent reviews. Particularly interesting is the case of integrable models in one dimension, where the dynamics is constrained by a large set of conserved charges [4], leading to peculiar features in the transport properties [5] or the relaxation towards a stationary state [6]. A key paradigm of closed-system dynamics is the quantum quench, where one typically prepares the pure ground state of a Hamiltonian which is then abruptly changed into a new one and the subsequent unitary time evolution is monitored [7]. The most common setting, well suited to study relaxation properties, is a global quench where the Hamiltonian is translational invariant both before and after the quench. However, if one is interested in transport properties far from equilibrium, some macroscopic inhomogeneity has to be present in the initial state.

In the context of spin chains, the simplest realization of such an inhomogeneity is a domain wall. In particular, for the XX chain a domain wall can be created by preparing the two halves of the system in their respective ground states with opposite magnetizations [8]. In the equivalent free-fermion representation, the simplest case of the maximally magnetized domain wall corresponds to a step-like initial condition for the occupation numbers. Under time evolution the initial inhomogeneity spreads ballistically, creating a front region which grows linearly in time. While the overall shape of the front is simple to obtain from a hydrodynamic (semi-classical) picture in terms of the fermionic excitations [9], the fine structure is more involved and shows universal features around the edge of the front [10,11]

The melting of domain walls has been considered in various different lattice models, such as the transverse Ising [12,13], the XY [14] and XXZ chains [15–18], hard-core bosons [19–21], as well as in the continuum for a Luttinger model [22], the Lieb-Liniger gas [23] or within conformal field theory [24,25]. Instead of a sharp domain wall, the melting of inhomogeneous interfaces can also be studied by applying a magnetic field gradient, which is then suddenly quenched to zero [26–28]. Mappings from the time-evolved state of an initial domain wall to the ground state of a specific Hamiltonian have also been established [26,29]. Very recently, domain-wall melting in disordered XXZ chains has been studied as a probe of many-body localization [30].

Here we consider another realization of a domain wall which is created in the ordered ferromagnetic phase of the transverse Ising chain. Starting from one of the symmetry-broken

ground states of the model, the order-parameter magnetization can be reversed along half of the chain. Due to the asymptotic degeneracy of the ordered states, this is still an eigenstate of the Ising chain locally, except for the neighbourhood of the kink in the magnetization where the domain-wall melting ensues.

The above setting has recently been studied numerically on infinite chains [31], using a matrix product state [32] related method, with two slightly different realization of the domain wall. For the excitation that is local in terms of Jordan-Wigner fermions, a very interesting observation on the magnetization profiles was made. Namely, it was pointed out that, after normalizing with the equilibrium value of the magnetization, the resulting snapshots of the profiles taken at times $ht$ (i.e. rescaled by the value of the transverse field $h$) all collapse onto each other to almost machine precision [31]. Furthermore, the universal profile was conjectured to be identical to the one [8] obtained for the free-fermion chain with the step-like initial state.

In this paper we revisit this problem and show that these features can be understood analytically. First, we show that a very simple semi-classical interpretation of the front profiles in the hydrodynamic scaling regime can be found. Moreover, in the limit of an infinite chain, even the fine structure of the profiles can be recovered by using a form-factor approach, providing an analytical support for the universality. For all of these results it turns out to be crucial that the domain-wall excitation is created by acting with a local fermion operator. Indeed, for a non-local realization of the same initial profile, the universality of the time-evolved front is lost and even the semi-classical picture breaks down.

The exact relation between the front profiles of the Ising and free-fermion domain-wall problems is quite remarkable. Indeed, in the latter case the time evolution is governed by a critical Hamiltonian whereas for the Ising chain we are always in the non-critical ferromagnetic regime. Despite the universal form of the magnetization profiles, one expects that this difference should clearly be reflected on the level of the time-evolved states. In fact, we will show that the entanglement entropy in the Ising chain always saturates for large times, in sharp contrast to the free-fermion case where it has a logarithmic growth in time [26,33,34]. Therefore, entanglement perfectly witnesses the non-criticality of the underlying Hamiltonian. Moreover, our results also indicate that the entropy in the non-equilibrium steady state of the Ising chain is equal to its ground-state value, suggesting that a non-trivial unitary transformation between these two states should exist.

The structure of the paper is as follows. In the next section we introduce the model and set up the basic formalism. The magnetization profiles for the Jordan-Wigner excitation are calculated in Sec. 3 using a number of different approaches. The results are then contrasted to those obtained for a non-local fermionic realization of the domain wall in Sec. 4. The time evolution of the entanglement entropy is discussed in Sec. 5 for both kinds of initial states. In Sec. 6 we show that some of the above results can naturally be carried over to the XY chain by duality. We conclude with a discussion of our results and their possible extensions in Sec. 7. The manuscript is supplemented by three appendices with various details of the analytical calculations.

## 2 Model and setting

We consider a finite transverse Ising (TI) chain of length $N$ with open boundaries, defined by the Hamiltonian

$$H_{TI} = -\frac{1}{2}\sum_{m=1}^{N-1}\sigma_m^x\sigma_{m+1}^x - \frac{h}{2}\sum_{m=1}^{N}\sigma_m^z. \tag{1}$$

The diagonalization of $H_{TI}$ follows standard practice by mapping the spins to fermions via a Jordan-Wigner transformation [35]. For the open chain it will be most convenient to work with Majorana fermions defined by

$$a_{2m-1} = \prod_{j=1}^{m-1} \sigma_j^z \, \sigma_m^x, \qquad a_{2m} = \prod_{j=1}^{m-1} \sigma_j^z \, \sigma_m^y, \tag{2}$$

and satisfying anticommutation relations $\{a_m, a_n\} = 2\delta_{m,n}$. The Jordan-Wigner transformation brings the Hamiltonian into a quadratic form in terms of the Majorana operators which can be further diagonalized via

$$\eta_k = \sum_{m=1}^{N} \frac{1}{2} \left[ \phi_k(m) \, a_{2m-1} - i\psi_k(m) \, a_{2m} \right]. \tag{3}$$

The $\eta_k$ are standard fermionic operators satisfying $\{\eta_k, \eta_l^\dagger\} = \delta_{k,l}$ and bring the Hamiltonian into the diagonal form

$$H = \sum_{k=1}^{N} \epsilon_k \eta_k^\dagger \eta_k + \text{const.} \tag{4}$$

The spectrum $\epsilon_k$ in Eq. (4) and the vectors $\phi_k$ and $\psi_k$ in Eq. (3) follow from the eigenvalue equations

$$(A-B)(A+B)\phi_k = \epsilon_k^2 \phi_k, \tag{5}$$
$$(A+B)(A-B)\psi_k = \epsilon_k^2 \psi_k, \tag{6}$$

that are solved numerically with the matrices

$$A_{mn} = \frac{1}{2}(\delta_{m,n-1} + \delta_{m,n+1}) - h\delta_{m,n}, \qquad B_{mn} = \frac{1}{2}(\delta_{m,n-1} - \delta_{m,n+1}). \tag{7}$$

We will now consider the ordered phase ($h < 1$) of the TI model. It is well known that one has an exponentially vanishing gap in the system size $N$, and the ground and first excited states become degenerate in the thermodynamic limit $N \to \infty$. For finite sizes, however, one has

$$|0\rangle = \frac{1}{\sqrt{2}}(|\Uparrow\rangle + |\Downarrow\rangle), \qquad |1\rangle = \frac{1}{\sqrt{2}}(|\Uparrow\rangle - |\Downarrow\rangle), \tag{8}$$

where $|\Uparrow\rangle$ and $|\Downarrow\rangle$ denote the macroscopically ordered states with a finite magnetization pointing in the $\pm x$ direction. Note, that for both $|0\rangle$ and $|1\rangle$ the magnetization vanishes since they respect the spin-flip symmetry of the Hamiltonian. The magnetization in the symmetry-broken ground states can thus be computed as

$$\langle\Uparrow|\sigma_n^x|\Uparrow\rangle = -\langle\Downarrow|\sigma_n^x|\Downarrow\rangle = \text{Re}\,\langle 0|\sigma_n^x|1\rangle. \tag{9}$$

Starting from the symmetry broken ground-state, we introduce two different types of initial states with a domain-wall magnetization profile

$$|\text{JW}\rangle = \prod_{j=1}^{n_0-1} \sigma_j^z \sigma_{n_0}^x |\Uparrow\rangle = a_{2n_0-1}|\Uparrow\rangle, \qquad |\text{DW}\rangle = \prod_{j=1}^{n_0-1} \sigma_j^z |\Uparrow\rangle = \prod_{j=1}^{n_0-1}(-ia_{2j-1}a_{2j})|\Uparrow\rangle. \tag{10}$$

Here JW stands for Jordan-Wigner, since the excitation is created by applying a single Majorana fermion, see (2). In contrast, DW is a simple domain-wall excitation which is, however, non-local in terms of the Majorana operators. It is easy to check that both of the above excitations simply flip the magnetization in the $x$-direction for all spins up to site $n_0 - 1$

$$\langle \text{JW}|\sigma_n^x|\text{JW}\rangle = \langle \text{DW}|\sigma_n^x|\text{DW}\rangle = \begin{cases} \langle\Uparrow|\sigma_n^x|\Uparrow\rangle & n \geq n_0 \\ \langle\Downarrow|\sigma_n^x|\Downarrow\rangle & n < n_0 \end{cases} \tag{11}$$

Additionally, the JW excitation creates a spin-flip in the $z$-direction at site $n_0$. To simplify the setting, we will consider a symmetric domain wall, $n_0 = N/2 + 1$ with $N$ even, in all of our numerical calculations. It should also be stressed that the domain wall is now in the longitudinal direction, in contrast to previous studies where a domain wall of the transverse magnetization was prepared.

The equilibrium magnetization can be computed by evaluating the matrix element in (9). Rewriting $\sigma_n^x$ with Majorana operators one has

$$\langle 0|\sigma_n^x|1\rangle = (-i)^{n-1}\langle 0|\prod_{j=1}^{2n-1} a_j \eta_1^\dagger|0\rangle \tag{12}$$

where we used $|1\rangle = \eta_1^\dagger|0\rangle$, corresponding to the the lowest-lying excitation with $\epsilon_1 \to 0$ for $N \gg 1$. Note that the vectors $\phi_1(m)$ and $\psi_1(m)$ defining the mode $\eta_1^\dagger$ are localized around the left/right boundary of the chain, with elements decaying exponentially on a characteristic boundary length scale $\xi_b \propto |\ln h|^{-1}$ [36,37].

We thus have to evaluate the expectation value of a string of Majorana operators in the ground state, which can be factorized, according to Wick's theorem, into products of two-point functions. The latter can be calculated as

$$\langle 0|a_j a_l|0\rangle = \delta_{j,l} + i\Gamma_{j,l}, \tag{13}$$

where the antisymmetric covariance matrix has a $2 \times 2$ block structure with matrix elements given by

$$\begin{aligned}\Gamma_{2m-1,2n} = -\Gamma_{2n,2m-1} = iG_{m,n}\\ \Gamma_{2m-1,2n-1} = \Gamma_{2m,2n} = 0\end{aligned} \quad , \qquad G_{m,n} = -\sum_k \phi_k(m)\psi_k(n). \tag{14}$$

One further needs the matrix elements with the edge mode

$$H_{2m-1} = \langle 0|a_{2m-1}\eta_1^\dagger|0\rangle = \phi_1(m), \qquad H_{2m} = \langle 0|a_{2m}\eta_1^\dagger|0\rangle = i\psi_1(m). \tag{15}$$

Finally, the magnetization at site $n$ can be written as a Pfaffian of a $2n \times 2n$ antisymmetric matrix [38,39]

$$\langle \Uparrow |\sigma_n^x| \Uparrow \rangle = \text{Pf}(M_0), \qquad M_0 = \begin{pmatrix} \Gamma & H \\ -H^T & 0 \end{pmatrix}. \tag{16}$$

Here $\Gamma$ denotes the $(2n-1) \times (2n-1)$ reduced covariance matrix, whereas $H$ (resp. its transpose) is a column (row) vector of length $2n-1$. The expression in (16) turns out to be real. Indeed, due to the simple checkerboard structure (14) of $\Gamma$, with nonvanishing elements only in the offdiagonals of the $2 \times 2$ blocks, the evaluation of the Pfaffian actually reduces to the calculation of the following $n \times n$ determinant

$$\langle \Uparrow |\sigma_n^x| \Uparrow \rangle = \begin{vmatrix} H_1 & G_{1,1} & G_{1,2} & \cdots & G_{1,n-1} \\ H_3 & G_{2,1} & G_{2,2} & \cdots & G_{2,n-1} \\ \vdots & \vdots & \vdots & \ddots & \vdots \\ H_{2n-1} & G_{n,1} & G_{n,2} & \cdots & G_{n,n-1} \end{vmatrix}. \tag{17}$$

## 3 Evolution of magnetization after Jordan-Wigner excitation

After having set up the basic formalism, we are now ready to consider the time evolution. First we deal with the JW excitation, where the time-evolved state reads

$$|\psi(t)\rangle = e^{-iH_{TI}t}|\text{JW}\rangle. \tag{18}$$

The most important observable we are interested in is the order parameter magnetization $\sigma_n^x$, for which results can be obtained using a number of different approaches. First, we follow along the lines of the previous section and derive analogous Pfaffian formulas for the time-evolved magnetization which are exact for open chains of finite size. The scaling behaviour of the results suggests that a simple interpretation within a semi-classical approach should exist, which is presented in the second subsection. To study the fine structure of the profile directly in the thermodynamic limit, $N \to \infty$, one has to follow a different route using the form-factor approach. At the end of the section we shortly discuss also the time evolution of the transverse magnetization $\sigma_n^z$.

## 3.1 Pfaffian approach

Instead of using the time-evolved state of Eq. (18), it is easier to work in a Heisenberg picture where the operators evolve as $\sigma_n^x(t) = e^{iH_{TI}t} \sigma_n^x e^{-iH_{TI}t}$. The time-dependent magnetization can then be obtained by taking expectation values in the initial state and can be written as

$$\langle JW| \sigma_n^x(t) |JW\rangle = \text{Re}\, \langle 0| a_{2n_0-1}(-i)^{n-1} \prod_{j=1}^{2n-1} a_j(t) a_{2n_0-1} \eta_1^\dagger |0\rangle. \tag{19}$$

The above formula is analogous to that of Eq. (12), one has, however, Heisenberg operators in the product surrounded by two extra $a_{2n_0-1}$ and thus one has to evaluate a string of $2n+2$ operators. In order to apply Wick's theorem, one first needs the time evolution of the Majorana operators

$$a_j(t) = e^{iH_{TI}t} a_j e^{-iH_{TI}t} = \sum_{l=1}^{2N} R_{jl} a_l \tag{20}$$

where the matrix elements of the propagator $R$ are given by

$$R_{2m-1,2n-1} = \sum_{k=1}^{N} \cos(\epsilon_k t) \phi_k(m) \phi_k(n), \qquad R_{2m,2n} = \sum_{k=1}^{N} \cos(\epsilon_k t) \psi_k(m) \psi_k(n),$$

$$R_{2m-1,2n} = -\sum_{k=1}^{N} \sin(\epsilon_k t) \phi_k(m) \psi_k(n), \qquad R_{2m,2n-1} = \sum_{k=1}^{N} \sin(\epsilon_k t) \psi_k(m) \phi_k(n). \tag{21}$$

It is easy to show that the two-point functions of the Heisenberg operators do not change in time, $\langle 0| a_j(t) a_l(t) |0\rangle = \langle 0| a_j a_l |0\rangle$. Indeed, since the Hamiltonian is unchanged in our protocol (i.e. there is no quench involved), the exponential factors in the Heisenberg operators act trivially on the ground state. However, the expectation values of products of operators at different times becomes nontrivial and, using (20) and (13), can be evaluated as

$$C_j = \langle 0| a_{2n_0-1} a_j(t) |0\rangle = R_{j,2n_0-1} - i \sum_{l=1}^{2N} R_{j,l} \Gamma_{l,2n_0-1},$$

$$D_j = \langle 0| a_j(t) a_{2n_0-1} |0\rangle = R_{j,2n_0-1} + i \sum_{l=1}^{2N} R_{j,l} \Gamma_{l,2n_0-1}. \tag{22}$$

The remaining two-point functions are given by $\langle 0| a_{2n_0-1} a_{2n_0-1} |0\rangle = 1$ and $\langle 0| a_{2n_0-1} \eta_1^\dagger |0\rangle = \phi_1(n_0)$.

With all the ingredients at hand, one can again arrange the two-point functions in an antisymmetric matrix $M$ of size $(2n+2) \times (2n+2)$ and calculate its Pfaffian. However, the

calculation can be simplified using the special properties of Pfaffians, as shown in detail in Appendix A. In turn, the result can be written in a form analogous to the equilibrium case

$$\langle \mathrm{JW} | \sigma_n^x(t) | \mathrm{JW} \rangle = -\mathrm{Re}\,\mathrm{Pf}(\tilde{M}), \qquad \tilde{M} = \begin{pmatrix} \tilde{\Gamma} & \tilde{H} \\ -\tilde{H}^T & 0 \end{pmatrix} \tag{23}$$

where $\tilde{M}$ is a matrix of size $2n \times 2n$ and its entries are given by

$$\tilde{\Gamma} = \Gamma - i(CD^T - DC^T), \qquad \tilde{H} = H - (C+D)\phi_1(n_0). \tag{24}$$

Here $\tilde{\Gamma}$ and $\tilde{H}$ are again a matrix of size $(2n-1) \times (2n-1)$ and a vector of length $2n-1$, respectively. However, due to the extra terms in Eq. (24), the matrix $\tilde{M}$ does not have the simple checkerboard structure as in equilibrium, and thus the result cannot be rewritten as a $n \times n$ determinant. Nevertheless, it is easy to show (see Appendix A) that all the elements of $\tilde{\Gamma}$ are real, and the only imaginary entries in $\tilde{H}$ are due to $H_{2m}$, see Eq. (15). Therefore, taking the real part in (23) is equivalent to setting $H_{2m} = 0$ and calculating a real-valued Pfaffian, which can be performed by efficient numerical algorithms [40].

The results for the magnetization are shown in Fig. 1 for a chain of length $N = 200$. One can observe a number of features from the unscaled profiles at fixed $t = 50$ (shown on the left). In particular, it is easy to see that the edges of the expanding front are located at a distance $\approx ht$ measured from the initial location $n_0 - 1/2 = (N+1)/2$ of the domain wall. From this it is easy to infer that the maximum speed of propagation is given by $v = h$ which will be verified by the semi-classical approach of the next subsection. Beyond the edge of the front one recovers, up to exponential tails, the equilibrium profile, which shows well-known boundary effects [36, 37] on a length scale $\xi_b$ close to the ends of the chain.

To better understand the behaviour of the front, one should compare snapshots of the magnetization, normalized by the equilibrium value, for various fields $h$ but keeping the scaling variable $ht = 50$ fixed, as depicted on the right of Fig. 1. Remarkably, as already noted in Ref. [31] for infinite chains, the data sets collapse close to machine precision on a single curve, which turns out to be identical to the one [8] for the free-fermion chain with a step-like initial state.

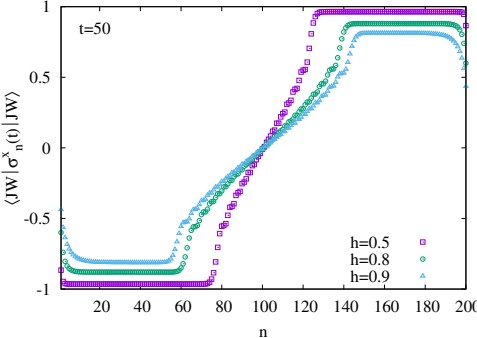 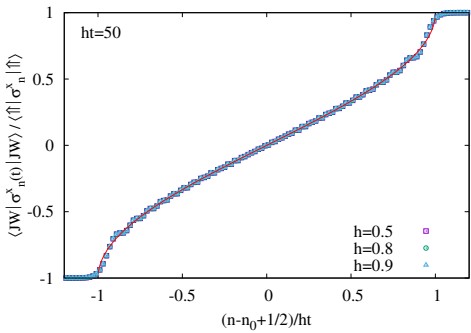

Figure 1: Magnetization profiles for a JW excitation in a chain of length $N = 200$. Left: at time $t = 50$ and for various values of $h$. Right: normalized profiles with a rescaled horizontal axis, for $ht = 50$ kept fixed. The symbols for various $h$ can not be distinguished due to the perfect data collapse. The solid line shows the semi-classical result, see Eqs. (26) and (32).

## 3.2 Semi-classical approach

To interpret the above results, we now present a very simple semi-classical argument which yields the correct magnetization profile for the JW excitation in the scaling regime, i.e. $|n -$

$n_0| \to \infty$ and $ht \to \infty$ with $(n - n_0)/ht$ kept fixed. To simplify the discussion, here we work directly with an infinite chain, with no boundary conditions imposed. Due to perfect translational invariance, the eigemodes created by $\eta_q^\dagger$ are now propagating waves with continuous momenta chosen from the interval $q \in [-\pi, \pi]$.

In the context of the TI chain, the semi-classical reasoning was originally presented by Sachdev and Young [41], and has since been used to obtain the magnetization for various (global or local) quench protocols [42, 43]. The argument is as follows: the initial Majorana operator $a_{2n_0-1}$ which excites the domain wall is, in fact, a mixture of the various eigenmodes excited by $\eta_q^\dagger$. One could think of each of these modes as an elementary domain-wall excitation which propagates at a given speed

$$v_q = \frac{d\epsilon_q}{dq} = \frac{h \sin q}{\epsilon_q}, \qquad \epsilon_q = \sqrt{(\cos q - h)^2 + \sin^2 q}, \tag{25}$$

with $\epsilon_q$ the dispersion of the TI chain. To get the magnetization at a given site $n$, one simply has to determine the density $\mathcal{N}$ of excitations that have sufficient velocities to arrive from the initial location $n_0$ to the point of observation in time $t$. Introducing the scaling variable $v = (n - n_0)/t$ and focusing on $v > 0$, one has

$$\langle \text{JW} | \sigma_n^x(t) | \text{JW} \rangle = \langle \Uparrow | \sigma_n^x | \Uparrow \rangle (1 - 2 \mathcal{N}(v)), \qquad \mathcal{N}(v) = \frac{q_+(v) - q_-(v)}{2\pi} \tag{26}$$

where the wave numbers $q_-(v) \le q \le q_+(v)$ satisfy $v_q \ge v$.

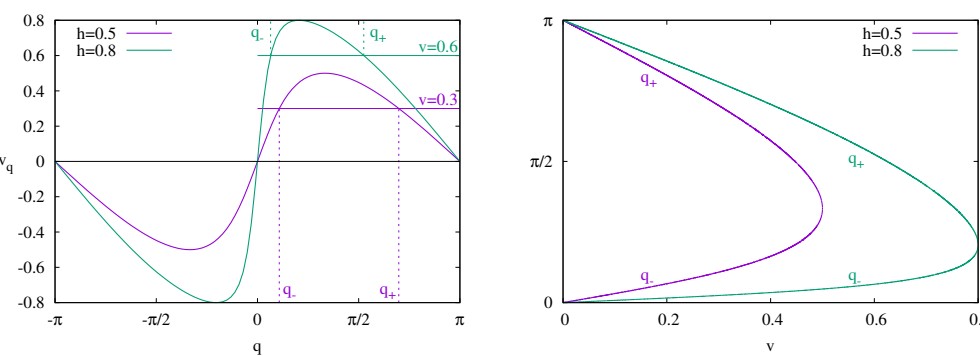

Figure 2: Left: graphical solution of the equation $v_q = v$. Right: wave numbers $q_-$ and $q_+$ as a function of $v$.

In other words, $\mathcal{N}(v)$ is just the fraction of domain-wall excitations that have a velocity larger than $v$. This can be obtained via (25), by solving the equation $v_{q_\pm} = v$ for $q_\pm(v)$, which is represented graphically on Fig. 2. The analytical solution can be found by introducing the variable $z = \sin q$, which leads to the following quadratic equation

$$h^2 z^2 = v^2 (1 - 2h\sqrt{1 - z^2} + h^2), \tag{27}$$

with the roots given by

$$h^2 z_\pm^2 = v^2 (1 + h^2) - 2v^4 \pm 2v^2 \sqrt{v^4 - v^2(1 + h^2) + h^2}. \tag{28}$$

Finally, the solution for the wavenumbers reads

$$q_- = \arcsin z_-, \qquad q_+ = \begin{cases} \arcsin z_+ & v \ge v_0 \\ \frac{\pi}{2} + \arccos z_+ & v < v_0 \end{cases}, \qquad v_0 = \frac{h}{\sqrt{1 + h^2}}, \tag{29}$$

where $v_0$ is the solution of the equation $z_+(v_0) = 1$.

Using the identity $\arcsin(z) + \arccos(z) = \pi/2$ and the addition formulas for the arccos function, the difference of the wavenumbers can be written as

$$q_+ - q_- = \begin{cases} \arccos\left(z_+ z_- - \sqrt{(1-z_+^2)(1-z_-^2)}\right) & v < v_0 \\ \arccos\left(z_+ z_- + \sqrt{(1-z_+^2)(1-z_-^2)}\right) & v \geq v_0 \end{cases}. \tag{30}$$

From the solutions (28) one finds

$$z_+ z_- = \frac{v^2}{h^2}(1-h^2), \qquad \sqrt{(1-z_+^2)(1-z_-^2)} = \left|1 - \frac{v^2}{h^2}(1+h^2)\right|. \tag{31}$$

It is easy to see that the right hand side term within the absolute value changes sign exactly at $v = v_0$, hence from (30) one finds that the minus sign applies for all values of $v$. Finally, substituting both terms one arrives at

$$\mathcal{N}(v) = \frac{q_+(v) - q_-(v)}{2\pi} = \frac{1}{2\pi}\arccos(2\frac{v^2}{h^2} - 1) = \frac{1}{\pi}\arccos\frac{v}{h}, \tag{32}$$

which is exactly the free-fermion result [8] with the velocity rescaled by $h$.

## 3.3 Form-factor approach

The semi-classical approach yields a very simple physical explanation for the magnetization profile in the scaling limit $|n - n_0| \to \infty$ and $t \to \infty$ with the ratio $v = |n - n_0|/t$ kept fixed. However, it does not account for the perfect collapse of the normalized magnetization curves $\langle JW|\sigma_n^x(t)|JW\rangle / \langle \Uparrow |\sigma_n^x| \Uparrow \rangle$ even at finite times for a fixed value of $ht$, see Fig. 1. To capture the fine-structure of the profile, one can follow a form-factor approach which was used successfully to obtain results for the magnetization in case of a global quench [44]. In contrast to the Pfaffian approach, which is well-suited for open chains of finite size, the form-factor approach works most efficiently in the thermodynamic limit.

Our starting assumption for the semi-classical approach was that the JW excitation is a mixture of the various single-particle eigenmodes. It turns out that, to make this statement rigorous, one has to consider a TI chain with antiperiodic boundary conditions $\sigma_{N+1}^x = -\sigma_1^x$. The Hamiltonian $H$ for the antiperiodic chain can be diagonalized by the very same procedure as the periodic one, which is summarized in Appendix B. The main feature of both geometries is that the Hilbert space splits up into the Neveu-Schwarz (NS) and Ramond (R) sectors, which differ by their symmetry properties with respect to a global spin-flip transformation. In particular, the vacua of the two sectors, $|0\rangle_{\text{NS}}$ and $|0\rangle_{\text{R}}$, are analogous to those $|0\rangle$ and $|1\rangle$ of the open chain in Eq. (8), and the symmetry-broken ground states are obtained as their linear combinations. In turn, the time-evolved magnetization after the JW excitation is given by

$$\langle JW|\sigma_n^x(t)|JW\rangle = \text{Re}\,{}_{\text{R}}\langle 0|a_{2n_0-1}e^{iHt}\sigma_n^x e^{-iHt}a_{2n_0-1}|0\rangle_{\text{NS}}. \tag{33}$$

The Jordan-Wigner excitation $a_{2n_0-1}$ can now be rewritten in the basis that diagonalizes the Hamiltonian, as shown in (84) of Appendix B. It will populate the vacua as

$$a_{2n_0-1}|0\rangle_{\text{NS}} = \frac{1}{\sqrt{N}}\sum_q e^{-iq(n_0-1)}e^{i\frac{\theta_q}{2}}|q\rangle_{\text{NS}}, \qquad {}_{\text{R}}\langle 0|a_{2n_0-1} = \frac{1}{\sqrt{N}}\sum_p {}_{\text{R}}\langle p|e^{ip(n_0-1)}e^{-i\frac{\theta_p}{2}}, \tag{34}$$

where the $\theta_q$ and $\theta_p$ are Bogoliubov angles defined in (81). The momenta $q$ of the NS sector (respectively $p$ of the R sector) are quantized differently: they are half-integer (integer) multiples of $2\pi/N$. Most importantly, the single-particle states $|q\rangle_{\text{NS}}$ and ${}_{\text{R}}\langle p|$ are exact eigenvectors of the antiperiodic Hamiltonian. Hence, their time evolution becomes trivial

$$e^{-iHt}|q\rangle_{\text{NS}} = e^{-i\epsilon_q t}|q\rangle_{\text{NS}}, \qquad {}_{\text{R}}\langle p|e^{iHt} = {}_{\text{R}}\langle p|e^{i\epsilon_p t}, \tag{35}$$

with the dispersion relation defined in (25). The role of the antiperiodic boundary conditions should be stressed at this point, since the eigenvectors of the periodic TI chain always have an even number of single-particle excitations.

Clearly, thanks to the simple time evolution in (35), the only remaining ingredients we need are the form factors $_\mathrm{R}\langle p|\sigma_n^x|q\rangle_\mathrm{NS}$ between the single-particle states. Fortunately, for the particular fermionic basis at hand, the form factors are known exactly and in the limit $N \to \infty$ are given by [45]

$$\frac{_\mathrm{R}\langle p|\sigma_n^x|q\rangle_\mathrm{NS}}{_\mathrm{R}\langle 0|\sigma_n^x|0\rangle_\mathrm{NS}} = -\frac{i}{N}\frac{\epsilon_p + \epsilon_q}{2\sqrt{\epsilon_p \epsilon_q}}\frac{e^{i(n-1/2)(q-p)}}{\sin\frac{q-p}{2}}. \tag{36}$$

The vacuum matrix element in the denominator of the left hand side is simply the equilibrium magnetization. In fact, the form factors for finite $N$ are also known exactly [45], but we are only interested in the thermodynamic limit. Combining the results (34)-(36) and turning the sums into integrals, one finally arrives at the result for the normalized magnetization

$$\frac{\langle \mathrm{JW}|\sigma_n^x(t)|\mathrm{JW}\rangle}{\langle \Uparrow |\sigma_n^x| \Uparrow\rangle} = \int_{-\pi}^{\pi}\frac{dp}{2\pi}\int_{-\pi}^{\pi}\frac{dq}{2\pi}\frac{\epsilon_p + \epsilon_q}{2\sqrt{\epsilon_p \epsilon_q}}\frac{\sin\left[(2(n-n_0)+1)\frac{q-p}{2}\right]}{\sin\frac{q-p}{2}}\cos\frac{\theta_q - \theta_p}{2}\cos(\epsilon_q - \epsilon_p)t. \tag{37}$$

To show the identity with the free-fermion result, one has to evaluate the above double integral. First, one notices that the Dirichlet kernel appears in the integrand of (37) which can be rewritten as

$$\frac{\sin\left[(2(n-n_0)+1)\frac{q-p}{2}\right]}{\sin\frac{q-p}{2}} = \sum_{k=-n+n_0}^{n-n_0}\cos k(q-p), \tag{38}$$

leaving us with a sum of integrals with simpler integrands to evaluate. Assuming that the result depends on the scaling variable $ht$ only (see Fig. 1), one can show after a rather tedious exercise, the details of which are given in Appendix C, that each of these integrals reproduce the square of a Bessel function

$$\int_{-\pi}^{\pi}\frac{dp}{2\pi}\int_{-\pi}^{\pi}\frac{dq}{2\pi}\frac{\epsilon_p + \epsilon_q}{2\sqrt{\epsilon_p \epsilon_q}}\cos k(q-p)\cos\frac{\theta_q - \theta_p}{2}\cos(\epsilon_q - \epsilon_p)t = J_k^2(ht). \tag{39}$$

Consequently, the normalized magnetization profile is obtained in the simple form

$$\frac{\langle \mathrm{JW}|\sigma_n^x(t)|\mathrm{JW}\rangle}{\langle \Uparrow |\sigma_n^x| \Uparrow\rangle} = \sum_{k=-n+n_0}^{n-n_0} J_k^2(ht), \tag{40}$$

which is indeed the free-fermion result of Ref. [8].

Finally, it should be pointed out that the semi-classical result (26) in the scaling limit, with the scaling function (32), could also be obtained from a saddle-point approximation of the double integral (37) along the lines of Ref. [46].

### 3.4 Transverse magnetization

To conclude this section, we shortly discuss the time-evolution of the transverse magnetization $\sigma_n^z$ for the open-chain geometry. As remarked earlier, the JW excitation flips the transverse spin at site $n_0$ and the disturbance spreads out in time. Since $\sigma_n^z$ is an even operator in terms of the fermions, it has only diagonal matrix elements w.r.t. the ground states $|0\rangle$ or $|1\rangle$, and these will coincide for $N \gg 1$. In turn one has

$$\langle \mathrm{JW}|\sigma_n^z(t)|\mathrm{JW}\rangle = -i\langle 0|a_{2n_0-1}a_{2n-1}(t)a_{2n}(t)a_{2n_0-1}|0\rangle = \tilde{\Gamma}_{2n-1,2n} \tag{41}$$

where the matrix element $\tilde{\Gamma}_{2n-1,2n}$ can be calculated according to Eq. (65) of Appendix A. In the limit $N \to \infty$, Eq. (41) can be shown to coincide with the corresponding result of Ref. [31].

The normalized profiles of the transverse magnetization are shown in Fig. 3. The spreading of the initially flipped spin can be seen on the left for $h = 0.5$. Interestingly, the total transverse magnetization seems to be conserved to a very good precision, at least until the front reaches the boundary region. On the right we show the profiles for various $h$ but with a fixed value of $ht = 25$. Clearly, in contrast to the order-parameter, the normalized transverse magnetization is not a function of $ht$ only.

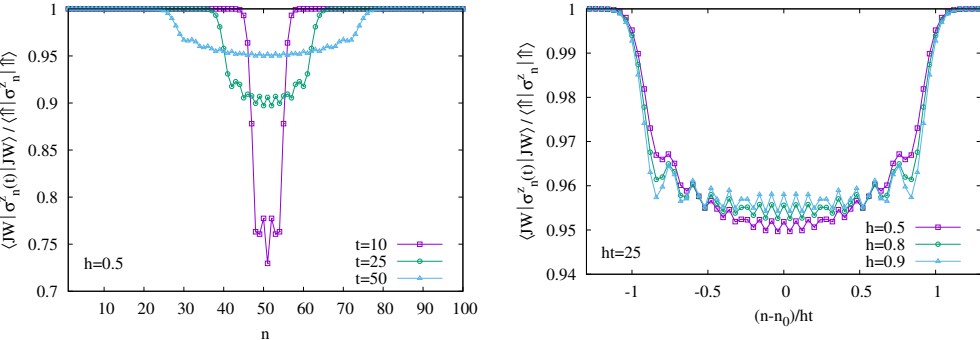

Figure 3: Normalized transverse magnetization profiles for JW excitation in a chain of length $N = 100$. Left: for $h = 0.5$ and various $t$. Right: for various $h$ with $ht = 25$ kept fixed, on a rescaled horizontal axis.

# 4 Magnetization profiles for domain-wall excitation

In the previous section we have shown that the profiles for the JW excitation can be obtained using various approaches and the underlying physics can be understood by a simple semi-classical argument. We now turn our attention towards the simple domain-wall excitation [31], defined on the right of Eq. (10). Although the difference from the JW excitation seems innocuous in the spin-representation, due to the non-locality of the Jordan-Wigner transformation, the DW excitation becomes a string of Majorana operators. Analogously to Eq. (19), the magnetization can now be written as

$$\langle \text{DW}|\sigma_n^x(t)|\text{DW}\rangle = \text{Re}\,(-1)^{n_0-1}(-i)^{n-1}\langle 0| \prod_{J=1}^{2n_0-2} a_J \prod_{j=1}^{2n-1} a_j(t) \prod_{J'=1}^{2n_0-2} a_{J'}\eta_1^{\dagger}|0\rangle. \qquad (42)$$

The expectation value in (42) can still be written as a Pfaffian, albeit with a matrix of much larger size. To this end we define the rectangular matrices $C$ and $D$ of unequal-time two-point functions with elements

$$\begin{aligned}
C_{j,J} &= \langle a_J a_j(t)\rangle = R_{j,J} - \sum_{k=1}^{2N} R_{j,k}\Gamma_{k,J} \\
D_{j,J} &= \langle a_j(t) a_J\rangle = R_{j,J} + \sum_{k=1}^{2N} R_{j,k}\Gamma_{k,J}
\end{aligned} \qquad (43)$$

where the capitalized index $J$ runs over the set $J = 1, \ldots, 2n_0 - 2$, whereas $j = 1, \ldots, 2n-1$ as before. Similarly, one can introduce the reduced covariance matrix $\Gamma_0$ (with elements $\Gamma_{J,J'}$)

and the column vector $H_0$ (with elements $H_J$) where again $J, J' = 1, \ldots, 2n_0 - 2$. Using these definitions, the magnetization can be written as a $(4n_0 - 4 + 2n) \times (4n_0 - 4 + 2n)$ Pfaffian, given explicitly in (67). Furthermore, performing manipulations similar to the JW case (see Appendix A), the expression can again be reduced to a Pfaffian of size $2n \times 2n$ given by

$$\langle \mathrm{DW} | \sigma_n^x(t) | \mathrm{DW} \rangle = \mathrm{Re}\,\mathrm{Pf}(\hat{M}), \qquad \hat{M} = \begin{pmatrix} \hat{\Gamma} & \hat{H} \\ -\hat{H}^T & 0 \end{pmatrix} \qquad (44)$$

where

$$\hat{\Gamma} = \Gamma - i(CD^T - DC^T) + (C+D)\Gamma_0(C+D)^T, \qquad \hat{H} = H - (C+D)H_0. \qquad (45)$$

The Pfaffian in (44) can be evaluated numerically with the results shown in Fig. 4. The normalized magnetization profiles are plotted against the distance from the initial location of the domain-wall, rescaled by $ht$. On the left of Fig. 4, we show the profiles at fixed $ht = 50$ and for various values of $h$. From the figure it becomes evident that the universality is lost as one finds no data collapse. Moreover, even the semi-classical result valid for the JW case and shown by the solid line, breaks down for the DW excitation: while the agreement for $h = 0.5$ still seems to be fairly good, the deviations increase dramatically when approaching the critical value of the field $h \to 1$.

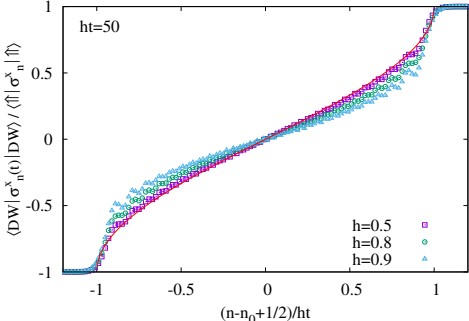
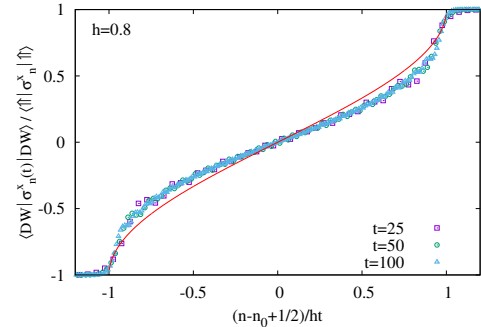

Figure 4: Normalized magnetization profiles vs. rescaled distance for DW excitation in a chain of length $N = 200$. Left: the profiles for fixed $ht = 50$ and various values of $h$ do not collapse. Right: the profiles for $h = 0.8$ and various times collapse onto each other. The solid lines show the semi-classical result, Eqs. (26) and (32), for the JW excitation for comparison.

The breakdown of the semi-classical picture does not come entirely unexpected. In fact, the DW initial state consists of a string of Majorana excitations extending over the left half-chain, which cannot any more be considered as a mixture of single-particle excitations in the momentum space. This becomes even more obvious in terms of the form-factor approach of the previous section. Indeed, in the DW case one has to consider many-particle form factors instead of the single-particle matrix elements of Eq. (36). Since these form factors become highly involved with increasing particle number [45], such a calculation is beyond our reach. Nevertheless, for a fixed value of $h$, one still has a ballistic expansion with the maximal signal velocity given by $h$, as demonstrated by the rescaled data on the right of Fig. 4.

Finally, one could also have a look at the transverse magnetization. Although, in contrast to the JW case, the initial state does not have any flipped spin in the $z$-direction, the profile will not remain constant. Indeed, one observes a signal front (albeit much weaker than in the JW case) propagating outwards from the location of the domain wall with the same speed $v = h$. In complete analogy with Eq. (41), the transverse magnetization for the DW case is given by $\langle \mathrm{DW} | \sigma_n^z(t) | \mathrm{DW} \rangle = \hat{\Gamma}_{2n-1,2n}$, with the corresponding matrix element defined in (70).

# 5   Entanglement evolution

Given the universal result (40) for the magnetization profile in the JW case, the question naturally emerges whether one has a deeper connection to the free-fermion domain-wall problem on the level of the time-evolved state. To answer this question, we shall now consider the entanglement entropy, which carries important information about the state itself. Entanglement evolution has been considered previously in Refs. [15, 26, 28, 31, 33, 34, 47] for various domain-wall initial states.

The time evolution of the entanglement entropy is given by $S(t) = -\text{Tr}\,\rho_A(t)\ln\rho_A(t)$ where $\rho_A(t) = \text{Tr}_B |\psi(t)\rangle\langle\psi(t)|$, with the state defined in Eq. (18). We consider various subsystem cuts and define $A = [1, n-1]$ and $B = [n, N]$. Changing the position of the left boundary $n = 2, \ldots, N$, we can determine the full entanglement profile and its time evolution along the chain. In particular, $n = n_0 = N/2 + 1$ corresponds to the half-chain entropy.

Even though there are several analytical approaches to obtain the entanglement entropy for Gaussian states of the TI chain (see e.g. Ref. [48] and references therein), the situation here is more subtle. Indeed, the initial state $|JW\rangle$ is defined in terms of the symmetry-broken ground state which, however, is not itself Gaussian but rather the superposition of two Gaussian states. Thus we will determine the entropy via density matrix renormalization group (DMRG) related calculations [49].

Within the matrix product state (MPS) formalism of DMRG [32], the ground state can be approximated to a very high precision by the ansatz

$$| \Uparrow \rangle \approx \sum_{s} A^{s_1} A^{s_2} \ldots A^{s_N} |s\rangle \,, \tag{46}$$

where $|s\rangle$ denotes the spin basis states and $A^{s_i}$ are auxiliary matrices with a variable bond dimension. They can be obtained by minimizing $\langle \Uparrow |H| \Uparrow \rangle$ with respect to the product $A^{s_i} A^{s_{i+1}}$ for a given site index $i$, while keeping all the other matrices fixed. Repeating the procedure for every pair of neighbouring lattice sites, the MPS will converge to the ground state after several sweeps. To ensure that we end up in the symmetry-broken ground state $| \Uparrow \rangle$, we introduced a small longitudinal field $h_x > 0$ in the Hamiltonian $H = H_{TI} - h_x \sum_i \sigma_i^x$ for the first few sweeps and set $h_x = 0$ afterwards, until convergence is reached. The JW and DW excitations can then be created by acting with their (trivial) matrix product operator representations on the ground-state MPS. Finally, the time-evolution of the states was implemented with the finite two-site time-dependent variational principle (TDVP) algorithm [50].

We start by looking at the initial entropy profile at time $t = 0$. One should point out that the state $|JW\rangle$ is created by acting on the symmetry-broken ground state by a product of strictly local (on-site) terms, which do not modify the entropy. One thus expects that the result for the real ground state $|0\rangle$ should be recovered, except for a $\ln 2$ contribution coming from the degeneracy, which is now removed. In the limit of an infinite chain $N \to \infty$, this is given via elliptic integrals by the analytical expression [51, 52]

$$S(0) = \frac{1}{12}\left[\ln\left(\frac{h^2}{16h'}\right) + (2 - h^2)\frac{2I(h)I(h')}{\pi}\right], \tag{47}$$

with $h' = \sqrt{1 - h^2}$, which we recover in the bulk ($\xi_b \ll n \ll N - \xi_b$) of the chain. For cuts within about a distance $\xi_b$ from the boundaries, the profile becomes inhomogeneous with increasing entropies for smaller subsystems.

The time evolution is depicted on Fig. 5. On the left we plot the entropy of a half-chain ($n = n_0$), with $S(0)$ subtracted, against the scaling variable $ht$. After a sudden increase, the curves show a slower, oscillatory approach towards an asymptotic value which seems to be given by $\ln 2$. Interestingly, the approach takes place from below, with the curves never crossing

the asymptotic value. The distance of the maxima is given by $ht = \pi$ to a good precision. Note, however, that the collapse of the curves against the variable $ht$ is good but not exact. On the right of Fig. 5 we show the full profiles for $h = 0.5$ and various times, again with $S(0)$ subtracted and with the distance $n - n_0$ of the cut from the centre rescaled by $ht$. The rescaled profiles converge towards a scaling function for large times, which remains unchanged for other values of $h$ (not shown on the figure) as well.

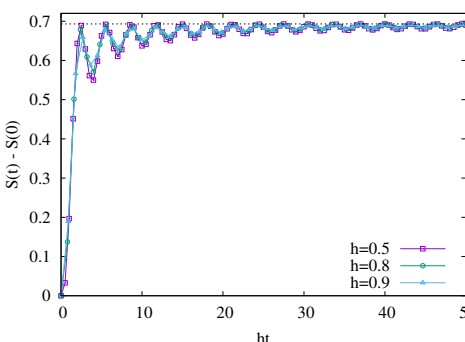 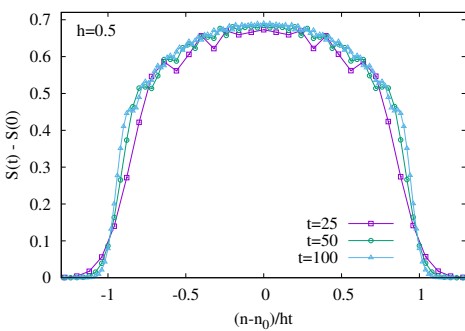

Figure 5: Time evolution of the entanglement entropy for a JW excitation in a chain of size $N = 200$, with the initial value $S(0)$ subtracted. Left: half-chain entropy vs. rescaled time for various values of $h$. The dotted line indicates the value $\ln 2$. Right: entropy profiles vs. rescaled distance from chain centre, for fixed h=0.5 and various times.

To interpret the above results, it is useful first to compare them to the corresponding result for the free-fermion case. There the entropy profile of an infinite chain is found to be given by the function [33, 34]

$$S = \frac{1}{6} \ln \left[ t(1 - v^2)^{3/2} \right] + \text{const.}, \qquad (48)$$

with the rescaled distance $v = (n - n_0)/t$ and $|v| < 1$. The latter profile is not only a function of $v$, but one has a contribution which grows logarithmically in time. Obviously, this is not the case for the JW excitation of the TI chain. In fact, the difference in the results gives perfect account about the underlying Hamiltonians: while for the free fermion the time-evolution is governed by a critical Hamiltonian, for the TI one is always in the $h < 1$ non-critical regime and thus the entropy saturates. It is important to stress that this difference is not revealed by looking only at the magnetization profiles, which are identical after rescaling.

Despite the difference in the entropy profiles, there is one important analogy which can be uncovered. We have observed (see left of Fig. 5) that the $t \to \infty$ result for the half-chain entropy is given by $S = S(0) + \ln 2$. However, as pointed out before, this is nothing else but the entropy of the real ground state $|0\rangle$. Moreover, from the scaling behaviour (see right of Fig. 5) one infers, that the same is true for arbitrary cuts with $|n - n_0|/ht \to 0$, i.e. finite distances from the centre and infinite time. This is exactly the regime, where a translational invariant current-carrying steady state is formed. Hence, no matter where we cut the system within the steady-state regime, we always get an entropy that is equal to the ground state value. Since the entropy gets contributions only from a distance of order $\xi$ of the correlation length measured from the cut, this suggests that the steady state, i.e. the reduced state of a *finite* segment of size $L \gg \xi$ in the limit $t \to \infty$, is unitarily equivalent to the reduced density matrix of the ground state

$$\lim_{t \to \infty} \lim_{N \to \infty} \text{Tr}_{N-L} |\psi(t)\rangle\langle\psi(t)| = U \left( \lim_{N \to \infty} \text{Tr}_{N-L} |0\rangle\langle0| \right) U^\dagger. \qquad (49)$$

In fact, this is exactly the case for the free-fermion chain, where the steady state is simply given by a boosted Fermi sea [28, 53]. Thus, taking a *finite* subsystem of length $L$ on the right-

hand side of site $n_0$ in the free-fermion chain, the asymptotic entropy for $t \gg L$ is given by the ground-state value $S = 1/3 \ln L + \text{const.}$, with the non-universal constant $\approx 0.726$. In this sense, the two results are completely analogous.

Finally, we study the entropy evolution also for the DW case. The results for the half-chain entropy as well as for the profiles are shown in Fig. 6. Although for smaller values of $h$ the half-chain entropy looks qualitatively similar to the JW case, for $h = 0.9$ there are noticeable differences. Namely, the increase for early times becomes slower, whereas for large times one has additional oscillations. Nevertheless, the asymptotical value of $S(t) - S(0)$ still seems to be given by $\ln 2$. Although the rescaled profiles collapse onto each other for fixed $h$ and various times, the shape of the scaling curves slightly changes for different values of $h$ in the DW case, as shown on the right of Fig. 6.

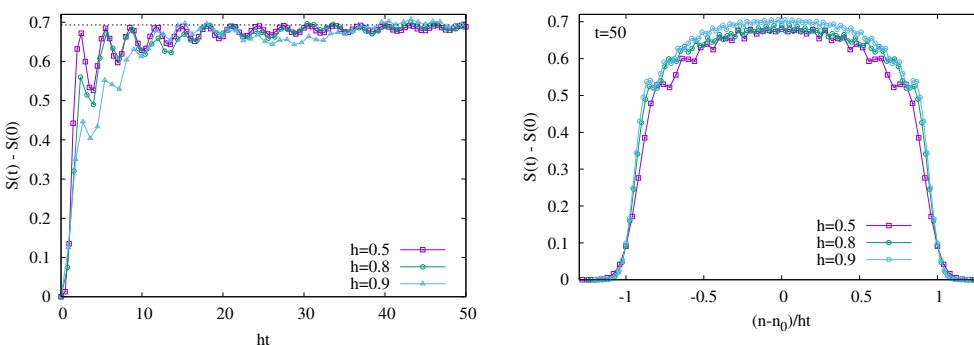

Figure 6: Time evolution of the entanglement entropy for a DW excitation in a chain of size $N = 200$, with the initial value $S(0)$ subtracted. Left: half-chain entropy vs. rescaled time for various values of $h$. The dotted line indicates the value $\ln 2$. Right: entropy profiles vs. rescaled distance from chain centre, for fixed $t = 50$ and various $h$.

# 6 Duality with the zero-field XY chain

It is natural to ask whether the results found for the magnetization and the entropy of the TI chain could exist for a broader universality class of spin models with ferromagnetic ground states. In the following we will show that the result naturally carries over to JW-type excitations of the anisotropic XY chain in zero magnetic field. Let us consider a chain of $2N$ sites defined by the Hamiltonian

$$H_{XY} = -\frac{1}{2} \sum_{n=1}^{2N-1} \left[ \frac{1+\gamma}{2} \sigma_n^x \sigma_{n+1}^x + \frac{1-\gamma}{2} \sigma_n^y \sigma_{n+1}^y \right]. \tag{50}$$

Applying the duality transformations [54–57]

$$\tau_i^{x,1} = \prod_{j=1}^{2i-1} \sigma_j^x, \qquad \tau_i^{x,2} = \prod_{j=1}^{2i-1} \sigma_j^y, \qquad \tau_i^{z,1} = \sigma_{2i-1}^y \sigma_{2i}^y, \qquad \tau_i^{z,2} = \sigma_{2i-1}^x \sigma_{2i}^x, \tag{51}$$

the XY Hamiltonian decomposes into the sum

$$H_{XY} = \frac{1+\gamma}{2} H_{TI,1} + \frac{1-\gamma}{2} H_{TI,2} \tag{52}$$

of two TI chains, defined in terms of the dual variables as

$$H_{TI,\alpha} = -\frac{1}{2} \sum_{i=1}^{N-1} \tau_i^{x,\alpha} \tau_{i+1}^{x,\alpha} - \frac{h_\alpha}{2} \sum_{i=1}^{N} \tau_i^{z,\alpha}, \qquad \alpha = 1, 2. \tag{53}$$

Here the magnetic fields are defined as

$$h_1 = \frac{1-\gamma}{1+\gamma}, \qquad h_2 = \frac{1+\gamma}{1-\gamma}. \tag{54}$$

Thus, the ground state of the XY Hamiltonian corresponds to the direct product of TI ground states on the corresponding sublattices. Note that, for $0 < \gamma < 1$, the Hamiltonian $H_{TI,1}$ is in its ordered phase, whereas $H_{TI,2}$ is in the disordered phase.

To calculate the magnetization for the XY chain, one has to rewrite the $\sigma^x$ operators in the dual variables

$$\sigma^x_{2i-1} = \prod_{j=1}^{i-1} \tau^{z,2}_j \tau^{x,1}_i, \qquad \sigma^x_{2i} = \prod_{j=1}^{i} \tau^{z,2}_j \tau^{x,1}_i. \tag{55}$$

Since both the ground states as well as the operators factorize on the two sublattices, one can write

$$\langle \Uparrow |\sigma^x_{2i-1}| \Uparrow \rangle_{XY} = \langle \Uparrow |\tau^{x,1}_i| \Uparrow \rangle_{TI,1} \langle 0| \prod_{j=1}^{i-1} \tau^{z,2}_j |0\rangle_{TI,2} \tag{56}$$

where we have used the fact that the lowest lying excitation of $XY$ corresponds to exciting the ordered $TI$ chain $H_{TI,1}$ only. The result for $\sigma^x_{2i}$ is similar. Furthermore, one can also construct the Majorana operators using the representation of the string variables in the dual language

$$\prod_{j=1}^{2n_0-2} \sigma^z_j \sigma^x_{2n_0-1} = \prod_{j=1}^{n_0-1} (-\tau^{z,1}_j \tau^{z,2}_j) \prod_{j=1}^{n_0-1} \tau^{z,2}_j \tau^{x,1}_{n_0} = \prod_{j=1}^{n_0-1} (-\tau^{z,1}_j) \tau^{x,1}_{n_0}. \tag{57}$$

In fact, this operator creates nothing else but a JW excitation of $H_{TI,1}$ (up to an irrelevant sign factor) while the ground state of $H_{TI,2}$ is left untouched

$$|\text{JW}\rangle_{XY} = \prod_{j=1}^{2n_0-2} \sigma^z_j \sigma^x_{2n_0-1}| \Uparrow \rangle_{XY} = (-1)^{n_0-1}|\text{JW}\rangle_{TI,1}|0\rangle_{TI,2}. \tag{58}$$

Finally, since the two TI Hamiltonians commute $\left[ H_{TI,1}, H_{TI,2} \right] = 0$, the time evolution operator also factorizes

$$\exp(-itH_{XY}) = \exp\left(-it\frac{1+\gamma}{2}H_{TI,1}\right) \exp\left(-it\frac{1-\gamma}{2}H_{TI,2}\right), \tag{59}$$

and one arrives at the relation

$$\frac{\langle \text{JW}|\sigma^x_{2i-1}(t)|\text{JW}\rangle_{XY}}{\langle \Uparrow |\sigma^x_{2i-1}| \Uparrow \rangle_{XY}} = \frac{\langle \text{JW}|\sigma^x_{2i}(t)|\text{JW}\rangle_{XY}}{\langle \Uparrow |\sigma^x_{2i}| \Uparrow \rangle_{XY}} = \frac{\langle \text{JW}|\tau^{x,1}_i(\frac{1+\gamma}{2}t)|\text{JW}\rangle_{TI,1}}{\langle \Uparrow |\tau^{x,1}_i| \Uparrow \rangle_{TI,1}}. \tag{60}$$

Hence, after proper rescaling, one indeed finds the universal free-fermion result (40) both on even and odd lattice sites. The relation (60) has also been checked against DMRG calculations with an excellent agreement.

The same argument also applies to the entanglement entropies and yields

$$S_{XY}(t) = S_{TI,1}\left(\frac{1+\gamma}{2}t\right) + S_{TI,2}(0). \tag{61}$$

Note that similar duality relations between entropies of $XY$ and $TI$ chains were found earlier for the ground state [58] as well as for local quenches [59].

## 7 Conclusions

We have studied the domain-wall melting for particular initial states of the ferromagnetic TI chain. For the JW excitation that is local in terms of the fermion operators that diagonalize the Hamiltonian, the longitudinal magnetization profiles after proper rescaling are completely identical to the ones observed for a fermionic hopping chain with step initial condition. The result carries over to the anisotropic XY chain with $h = 0$. The entanglement entropy is, however, found to saturate during time evolution and signals the non-criticality of the underlying Hamiltonian.

The case of the non-local DW excitation is quite different. In particular, the semi-classical approach, that yields the correct JW profiles in the scaling limit, breaks down and thus we have not been able to find an analytical result for the DW profiles. It might be possible to derive some results via the form factor approach which, however, also becomes highly involved and we have thus left this question open for future studies.

There are also a number of natural extensions of this work. First of all, one should check if the universality of the JW magnetization profiles extends to the full ferromagnetic phase of the XY model. A further step would be to investigate more general spin chains, such as the XXZ chain, that cannot be transformed into free fermions. While we do not expect the full universality for the fine structure of the profile to hold in this case, some essential features might still be inherited. It would also be instructive to compare the results to a quench setting, where the $|\Uparrow\rangle$ and $|\Downarrow\rangle$ states are prepared as the symmetry-broken ground states of two half chains which are then joined together.

Finally, our results for the entropy lead us to the conjecture that the non-equilibrium steady state is locally (i.e. in the region where the front has already swept through) related to the symmetry unbroken ground state of the TI chain. It would be interesting to find further evidence by comparing more complicated observables, such as spin correlation functions, which could also be obtained from the Pfaffian formalism.

## Acknowledgments

We thank P. Calabrese, A. Gambassi, M. Kormos and V. Zauner-Stauber for useful discussions. The authors acknowledge funding from the Austrian Science Fund (FWF) through Lise Meitner Project No. M1854-N36, and through SFB ViCoM F41, project P04. This research was supported in part by the National Science Foundation under Grant No. NSF PHY-1125915.

## A  Manipulations with Pfaffians

In this appendix we present the main steps that are needed to derive the results for the magnetization in Eqs. (23) and (44). We start by listing the most important properties of Pfaffians:

- Multiplication of a row and a column by a constant is equivalent to multiplication of the Pfaffian by the same constant.

- Simultaneous interchange of two different rows and corresponding columns changes the sign of the Pfaffian.

- A multiple of a row and corresponding column added to another row and corresponding column does not change the value of the Pfaffian.

- For a $2n \times 2n$ antisymmetric matrix $M$ and constant $\lambda$ one has $\mathrm{Pf}(\lambda M) = \lambda^n \mathrm{Pf}(M)$

- The Pfaffian of a $2n \times 2n$ antisymmetric matrix $M$ can be expanded into minors according to the reduction rule

$$\text{Pf}(M) = \sum_{j=2}^{2n} (-1)^j M_{1,j} \text{Pf}(M_{(1,j)}) \tag{62}$$

where $M_{(1,j)}$ is a $(2n-2) \times (2n-2)$ antisymmetric matrix obtained by removing the first and $j$-th rows and columns of $M$.

The above rules are very similar to the properties of determinants, except that one has to manipulate the rows and columns simultaneously.

## A.1 JW excitation

We first deal with the simpler JW excitation. According to (19), the magnetization is given by the expectation value of a string of $2n+2$ Majorana operators. Hence, it can be rewritten as the following Pfaffian

$$\langle \text{JW} | \sigma_n^x(t) | \text{JW} \rangle = \text{Re} \left[ (-i)^{n-1} \text{Pf}(M) \right], \qquad M = \begin{pmatrix} 0 & C^T & 1 & \phi_1(n_0) \\ -C & i\Gamma & D & H \\ -1 & -D^T & 0 & \phi_1(n_0) \\ -\phi_1(n_0) & -H^T & -\phi_1(n_0) & 0 \end{pmatrix}, \tag{63}$$

where we used a block-notation with $(2n-1) \times (2n-1)$ matrix $\Gamma$ and column-vectors $H$, $C$ and $D$ of length $2n-1$ defined in (15) and (22). Note that the transpose of the above vectors simply give the corresponding row-vectors. The remaining entries correspond to the expectation values $\langle 0 | a_{2n_0-1} a_{2n_0-1} | 0 \rangle = 1$ and $\langle 0 | a_{2n_0-1} \eta_1^\dagger | 0 \rangle = \phi_1(n_0)$.

We can now use the Pfaffian rules above to transform the matrix $M$ into matrices of simpler structure $M'$ and $M''$ given by

$$M' = \begin{pmatrix} 0 & C^T + D^T & 1 & 0 \\ -(C+D) & i\Gamma & D & H \\ -1 & -D^T & 0 & \phi_1(n_0) \\ 0 & -H^T & -\phi_1(n_0) & 0 \end{pmatrix},$$

$$M'' = \begin{pmatrix} 0 & 0 & 1 & 0 \\ 0 & i\tilde{\Gamma} & D & \tilde{H} \\ -1 & -D^T & 0 & \phi_1(n_0) \\ 0 & -\tilde{H}^T & -\phi_1(n_0) & 0 \end{pmatrix}. \tag{64}$$

In the first step, we have subtracted the third row/column of $M$ from the first ones which yields $M'$ on the left of Eq. (64). Subsequently, one can subtract the third row (column) of $M'$ multiplied by $C + D$ (respectively $C^T + D^T$) from the second row (column) which then leads to $M''$ on the right of Eq. (64), with $\tilde{\Gamma}$ and $\tilde{H}$ given in (24) of the main text. Clearly there is now only one nonzero entry in the first row/column of $M''$, it can thus be reduced to a smaller matrix of size $2n \times 2n$ by removing the first and third rows and columns. Indeed, using Eq. (62), the only nonvanishing contribution is with $j = 2n + 1$ which gives and extra sign for the reduced Pfaffian. Finally, the factor $(-i)^{n-1}$ in (63) can be absorbed by multiplying all the matrix elements by $-i$, except for the last row and column. This yields the final result in Eq. (23).

It is instructive to write out explicitly the matrix elements of $\tilde{\Gamma}$ and $\tilde{H}$

$$\tilde{\Gamma}_{i,j} = \Gamma_{i,j} + 2R_{i,2n_0-1}\sum_{l=1}^{2N}R_{j,l}\Gamma_{l,2n_0-1} - 2R_{j,2n_0-1}\sum_{l=1}^{2N}R_{i,l}\Gamma_{l,2n_0-1},$$
$$\tilde{H}_j = H_j - 2R_{j,2n_0-1}\phi_1(n_0). \tag{65}$$

Note that all the entries $\tilde{\Gamma}_{i,j}$ are real, and the only imaginary entries in $\tilde{H}_j$ appear for $j = 2n$ due to $H_{2n} = i\psi(n)$. In particular, for $t = 0$ the propagator $R_{i,j} = \delta_{i,j}$ is given by the identity and one has

$$\tilde{\Gamma}_{i,j} = \Gamma_{i,j} - 2\delta_{i,2n_0-1}\Gamma_{2n_0-1,j} - 2\delta_{j,2n_0-1}\Gamma_{i,2n_0-1}, \qquad \tilde{H}_j = H_j - 2\delta_{j,2n_0-1}\phi_1(n_0). \tag{66}$$

Now, if $n < n_0$, the extra terms in (66) do not give any contribution such that the $\tilde{M} = M_0$ and so the magnetization $-\mathrm{Re}\,\mathrm{Pf}(\tilde{M})$ is given by $-1$ times the equilibrium one. On the other hand, for $n \geq n_0$, the extra contributions simply reverse the sign of the $2n_0 - 1$-th row and column of $M_0$, giving an extra sign and reproducing the equilibrium magnetization.

## A.2 DW excitation

The case of the DW excitation is slightly more complicated since the magnetization (42) is given by a longer string of size $4n_0 - 4 + 2n$. Hence, it can be written as a Pfaffian of a $(4n_0 - 4 + 2n) \times (4n_0 - 4 + 2n)$ matrix

$$\langle \mathrm{DW}|\sigma_n^x(t)|\mathrm{DW}\rangle = \mathrm{Re}\left[(-1)^{n_0-1}(-i)^{n-1}\mathrm{Pf}(M)\right],$$
$$M = \begin{pmatrix} i\Gamma_0 & C^T & \mathbb{1} + i\Gamma_0 & H_0 \\ -C & i\Gamma & D & H \\ -\mathbb{1} + i\Gamma_0 & -D^T & i\Gamma_0 & H_0 \\ -H_0^T & -H^T & -H_0^T & 0 \end{pmatrix}, \tag{67}$$

where we used again block notation with square reduced covariance matrix $\Gamma_0$ and identity $\mathbb{1}$ of size $(2n_0 - 2) \times (2n_0 - 2)$, column vector $H_0$ of length $2n_0 - 2$ and rectangular matrices $C$ and $D$ of size $(2n - 1) \times (2n_0 - 2)$ defined in (43).

We will again manipulate the matrix $M$ and transform it to simpler forms $M'$ and $M''$ given by

$$M' = \begin{pmatrix} 0 & C^T + D^T & \mathbb{1} & 0 \\ -(C+D) & i\Gamma & D' & H \\ -\mathbb{1} & -D'^T & 0 & H_0 \\ 0 & -H^T & -H_0^T & 0 \end{pmatrix}, \qquad M'' = \begin{pmatrix} 0 & 0 & \mathbb{1} & 0 \\ 0 & i\hat{\Gamma} & D' & \hat{H} \\ -\mathbb{1} & -D'^T & 0 & H_0 \\ 0 & -\hat{H}^T & -H_0^T & 0 \end{pmatrix}. \tag{68}$$

In the first step, we do a row-by-row (resp. column-by-column) subtraction of the matrices in the third row (column) from the first ones in the block matrix $M$. This zeroes out the entries $i\Gamma_0$ and $H_0$ in the first row/column and transforms $-C \to -(C+D)$ (resp. $C^T \to C^T + D^T$). The remaining $\pm\mathbb{1}$ can be used to cancel out the $i\Gamma_0$ matrix in the third diagonal entry of $M$, by subtracting $i\Gamma_0/2$ (resp. its transpose) times the first row/column from the third ones. This yields $M'$ of Eq. (68) with a modified rectangular matrix defined as

$$D' = D - \frac{1}{2}(C+D)i\Gamma_0. \tag{69}$$

In the next step, we can cancel out the remaining entries $C^T + D^T$ and its transpose from the first row and column by subtracting the respective multiple of the third column/row from the second ones, which leads to $M''$ in Eq. (68) with $\hat{\Gamma}$ and $\hat{H}$ defined in (45).

Now, we can continue with the reduction of the matrix. The $\pm 1$ in the first row/column shows that one can eliminate $2 \times (2n_0 - 2)$ rows/columns consecutively, reducing again the matrix to a size of $2n \times 2n$. According to (62), every second step in the reduction gives a sign, which amounts to a factor $(-1)^{n_0-1}$ and cancels out with the respective sign term in (67). Finally, the $(-i)^{n-1}$ can again be absorbed just like in case of the JW calculation, and leads to the result in Eq. (44) in the main text.

One can again have a look at the matrix elements $\hat{\Gamma}_{i,j}$ and $\hat{H}_j$. Evaluating the matrix products in (45), one is left with the following simple expression

$$\hat{\Gamma}_{i,j} = \Gamma_{i,j} - 2 \sum_{J,\bar{J}} (R_{i,J} \Gamma_{J,\bar{J}} R_{j,\bar{J}} + R_{i,\bar{J}} \Gamma_{\bar{J},J} R_{j,J}), \qquad \hat{H}_j = H_j - 2 \sum_J R_{j,J} H_J \qquad (70)$$

where the sum over $J$ runs on the index set $J = 1, \ldots, 2n_0 - 2$ whereas the sum over $\bar{J}$ runs on the complement set $\bar{J} = 2n_0 - 1, \ldots, 2N$. It is easy to check how this again gives the correct result for $t = 0$, where $R_{i,j} = \delta_{i,j}$. Indeed, setting $n < n_0$, then since $i, j \leq 2n-1$ one has $R_{i,\bar{J}} = 0$ and $R_{j,\bar{J}} = 0$ for all $i, j$ and thus $\hat{\Gamma} = \Gamma$. However, $\hat{H} = -H$ and thus the last row/column of the Pfaffian is multiplied by $-1$ which changes its sign and thus the magnetization is given by $-\mathrm{Pf}(M_0)$. On the other hand, for $n \geq n_0$ some of the matrix elements of $\hat{\Gamma}$ will be changed. Indeed, one has

$$\hat{\Gamma}_{i,j} = \begin{cases} \Gamma_{i,j} & \text{if } i, j \leq 2n_0 - 2 \text{ or } i, j > 2n_0 - 2 \\ -\Gamma_{i,j} & \text{if } i \leq 2n_0 - 2, j > 2n_0 - 2 \text{ or } i > 2n_0 - 2, j \leq 2n_0 - 2 \end{cases},$$

$$\hat{H}_j = \begin{cases} -H_j & \text{if } j \leq 2n_0 - 2 \\ H_j & \text{if } j > 2n_0 - 2 \end{cases}. \qquad (71)$$

The above transformation simply amounts to multiplying all the columns/rows between $2n_0 - 1$ and $2n$ of the Pfaffian, each of which giving a sign. However, since there are an even number of rows and columns involved, in the end the value of the Pfaffian is unchanged and we get back the correct result $\mathrm{Pf}(M_0)$ for the magnetization.

## B  Diagonalization of $H_{TI}$ with antiperiodic boundary conditions

The TI chain with antiperiodic boundary conditions is given by the same Hamiltonian as in Eq. (1), except that both sums run until $m = N$ and we set $\sigma_{N+1}^x = -\sigma_1^x$. To diagonalize it, we follow a slightly different route along the lines of Ref. [45]. Instead of working with Majorana fermions, we define creation/annihilation operators

$$c_n = \prod_{j=1}^{n-1} \sigma_j^z \sigma_n^-, \qquad c_n^\dagger = \prod_{j=1}^{n-1} \sigma_j^z \sigma_n^+, \qquad (72)$$

where $\sigma_n^\pm = (\sigma_n^x \pm i\sigma_n^y)/2$ and the commutation relations are given by $\{c_m, c_n^\dagger\} = \delta_{m,n}$. We also introduce the global spin-flip operator

$$W = \prod_{j=1}^N \sigma_j^z = \prod_{j=1}^N (2c_j^\dagger c_j - 1), \qquad (73)$$

which commutes with the Hamiltonian $[H, W] = 0$. In terms of the fermion operators it reads

$$H = -\frac{1}{2} \sum_{n=1}^N \left[ (c_{n+1}^\dagger + c_{n+1})(c_n^\dagger - c_n) + h(2c_n^\dagger c_n - 1) \right], \qquad (74)$$

and the boundary condition for the fermions becomes $c_{N+1} = W c_1$. Since $W^2 = 1$, the eigenstates of the Hamiltonian split up into two sectors: the Ramond (R) sector corresponding to eigenvalue $W = 1$ has periodic, whereas the Neveu-Schwarz (NS) sector with $W = -1$ has antiperiodic boundary conditions for the fermions.

For our purposes it will be more convenient to work in a dual basis defined by

$$c_{n+1}^\dagger + c_{n+1} = d_n^\dagger + d_n, \qquad c_n^\dagger - c_n = d_n^\dagger - d_n. \tag{75}$$

The dual transformation interchanges the two terms in the Hamiltonian

$$H = \frac{1}{2} \sum_{n=1}^{N} \left[ (2d_n^\dagger d_n - 1) - h(d_{n+1}^\dagger - d_{n+1})(d_n^\dagger + d_n) \right], \tag{76}$$

where the dual fermions satisfy $\{d_m, d_n^\dagger\} = \delta_{m,n}$ and the same boundary condition $d_{N+1} = W d_1$. One then introduces the Fourier modes

$$d_{q_k} = \frac{1}{\sqrt{N}} \sum_{n=1}^{N} e^{-iq_k n} d_n, \tag{77}$$

where the momenta are quantized depending on which sector of the Hilbert space one chooses

$$q_k = \begin{cases} \frac{2\pi k}{N} & \text{if } W = 1 \text{ (R)} \\ \frac{2\pi(k+1/2)}{N} & \text{if } W = -1 \text{ (NS)} \end{cases}, \qquad k = -\frac{N}{2}, \dots, \frac{N}{2} - 1. \tag{78}$$

In terms of the Fourier modes, (76) can be rewritten as

$$H = \frac{1}{2} \sum_q \left[ (2d_q^\dagger d_q - 1)(1 - h \cos q) + i(d_q^\dagger d_{-q}^\dagger + d_q d_{-q}) h \sin q \right], \tag{79}$$

where the summation goes over the momenta defined by (78), but we omitted the $k$ indices for notational simplicity. The above Hamiltonian can be diagonalized by a Bogoliubov transformation

$$b_q = \cos(\theta_q/2) d_q + i \sin(\theta_q/2) d_{-q}^\dagger, \qquad b_{-q}^\dagger = \cos(\theta_q/2) d_{-q}^\dagger + i \sin(\theta_q/2) d_q, \tag{80}$$

where the dual Bogoliubov angle is given by

$$\tan \theta_q = \frac{h \sin q}{1 - h \cos q}. \tag{81}$$

The diagonal form of the Hamiltonian and the one-particle spectrum read

$$H = \sum_q \epsilon_q b_q^\dagger b_q, \qquad \epsilon_q = \sqrt{1 + h^2 - 2h \cos q}. \tag{82}$$

The many-particle eigenstates of the antiperiodic Hamiltonian can then be constructed as

$$|q_1, q_2, \dots, q_{2m+1}\rangle_{\text{NS}} = b_{q_1}^\dagger b_{q_2}^\dagger \dots b_{q_{2m+1}}^\dagger |0\rangle_{\text{NS}}, \qquad |p_1, p_2, \dots, p_{2n+1}\rangle_{\text{R}} = b_{p_1}^\dagger b_{p_2}^\dagger \dots b_{p_{2n+1}}^\dagger |0\rangle_{\text{R}}. \tag{83}$$

In fact, all the eigenstates have an odd number of excitations, as opposed to the periodic chain where the number of excitations is always even.

Finally, it is useful to rewrite the Majorana fermions of section 2 in terms of the eigenmodes of the Hamiltonian

$$a_{2n-1} = c_n + c_n^\dagger = \frac{1}{\sqrt{N}} \sum_q e^{-iq(n-1)}(d_q^\dagger + d_{-q}) = \frac{1}{\sqrt{N}} \sum_q e^{-iq(n-1)} e^{i\theta_q/2} (b_q^\dagger + b_{-q}), \tag{84}$$

which then leads directly to Eq. (34) in the main text.

# C  Integral formulas

In this appendix we will evaluate the integral

$$I_k = \int_{-\pi}^{\pi} \frac{dp}{2\pi} \int_{-\pi}^{\pi} \frac{dq}{2\pi} \frac{\epsilon_p + \epsilon_q}{2\sqrt{\epsilon_p \epsilon_q}} \cos k(q-p) \cos \frac{\theta_q - \theta_p}{2} \cos(\epsilon_q - \epsilon_p)t . \tag{85}$$

The factors involving the Bogoliubov angle can be written for $q > 0$ as

$$\cos \frac{\theta_q}{2} = \sqrt{\frac{(1+\epsilon_q)^2 - h^2}{4\epsilon_q}}, \qquad \sin \frac{\theta_q}{2} = \sqrt{\frac{h^2 - (1-\epsilon_q)^2}{4\epsilon_q}} . \tag{86}$$

First we will consider the simplest case $k = 0$. The integral then simplifies to

$$I_0 = \int_0^{\pi} \frac{dp}{\pi} \int_0^{\pi} \frac{dq}{\pi} \sqrt{\frac{\epsilon_p}{\epsilon_q}} \cos \frac{\theta_q}{2} \cos \frac{\theta_p}{2} \cos(\epsilon_q - \epsilon_p)t , \tag{87}$$

where we made use of the symmetry under exchange of $p$ and $q$ and the fact that the similar integral with $\sin \frac{\theta_q}{2} \sin \frac{\theta_p}{2}$ vanishes due its oddness under reflections $\theta_{-q} = -\theta_q$ or $\theta_{-p} = -\theta_p$.

To evaluate (87) it is more convenient to introduce $\epsilon_q = 1 + h\tilde{\epsilon}_q$ (similarly for $\epsilon_p$) and rewrite the integral in terms of the $\tilde{\epsilon}$ variables. The change of the integration measure can be derived from

$$\frac{d\tilde{\epsilon}_q}{dq} = \frac{1}{h} \frac{d\epsilon_q}{dq} = \frac{\sin q}{\epsilon_q} = \frac{\sqrt{1 - \left[\frac{h}{2}(1 - \tilde{\epsilon}_q^2) - \tilde{\epsilon}_q\right]^2}}{1 + h\tilde{\epsilon}_q} . \tag{88}$$

In terms of the new variables the integral reads

$$\int_{-1}^{1} \frac{d\tilde{\epsilon}_p}{\pi} \int_{-1}^{1} \frac{d\tilde{\epsilon}_q}{\pi} \frac{1}{\sqrt{1 - \tilde{\epsilon}_q^2}} \frac{(1 + h\tilde{\epsilon}_p)}{\sqrt{1 - \tilde{\epsilon}_p^2}} \cos(\tilde{\epsilon}_q - \tilde{\epsilon}_p)ht . \tag{89}$$

Now we can use the following integral formulas

$$\int_{-1}^{1} \frac{d\tilde{\epsilon}}{\pi} \frac{\cos(\tilde{\epsilon}ht)}{\sqrt{1 - \tilde{\epsilon}^2}} = J_0(ht), \qquad \int_{-1}^{1} \frac{d\tilde{\epsilon}}{\pi} \frac{\tilde{\epsilon}\cos(\tilde{\epsilon}ht)}{\sqrt{1 - \tilde{\epsilon}^2}} = \int_{-1}^{1} \frac{d\tilde{\epsilon}}{\pi} \frac{\sin(\tilde{\epsilon}ht)}{\sqrt{1 - \tilde{\epsilon}^2}} = 0 \tag{90}$$

to arrive at the result $I_0 = J_0^2(ht)$.

Unfortunately, the treatment of the general case $k > 0$ is much more cumbersome. On one hand, there are no simplifications due to symmetries of the integrand and thus one has many more terms appearing. On the other hand, even though the transformation to the $\tilde{\epsilon}$ variables yields the natural scaling variable $ht$ in the argument of the time-dependent cosine in (85), it also transforms the term $\cos k(q-p)$ to a more complicated expression. Indeed, using trigonometric identities, the extra factors can be rewritten in terms of the Chebyshev polynomials

$$\cos kq = T_k(\cos q), \qquad \sin kq = \sin q\, U_{k-1}(\cos q) \tag{91}$$

where, however, the argument has to be reexpressed as

$$z_q = \cos q = \frac{h}{2}(1 - \tilde{\epsilon}_q^2) - \tilde{\epsilon}_q \tag{92}$$

and similarly for $p$. Applying trigonometric addition formulas in the other cosine terms as well, the integral splits into a number of terms

$$I_k = I_{1,k}\hat{I}_{1,k} + I_{2,k}\hat{I}_{2,k} + I_{3,k}\hat{I}_{3,k} + I_{4,k}\hat{I}_{4,k} , \tag{93}$$

where we defined

$$I_{1,k} = \int_{-1}^{1} \frac{d\tilde{\epsilon}}{\pi} T_k(z) \frac{\cos(\tilde{\epsilon}ht)}{\sqrt{1-\tilde{\epsilon}^2}}, \qquad\qquad I_{2,k} = \int_{-1}^{1} \frac{d\tilde{\epsilon}}{\pi} T_k(z) \frac{\sin(\tilde{\epsilon}ht)}{\sqrt{1-\tilde{\epsilon}^2}}, \qquad (94)$$

$$I_{3,k} = \int_{-1}^{1} \frac{d\tilde{\epsilon}}{\pi} U_{k-1}(z) \frac{h}{2}\sqrt{1-\tilde{\epsilon}^2}\cos(\tilde{\epsilon}ht), \qquad I_{4,k} = \int_{-1}^{1} \frac{d\tilde{\epsilon}}{\pi} U_{k-1}(z) \frac{h}{2}\sqrt{1-\tilde{\epsilon}^2}\sin(\tilde{\epsilon}ht). \quad (95)$$

The integrals with the hat symbols are very similar to the ones defined above, but with an additional factor $(1+h\tilde{\epsilon})$ in the integrand, analogously to (89). Note that we used the shorthand notation $z$, defined in Eq. (92), in the arguments of the Chebyshev polynomials to simplify formulas.

The exact evaluation of the above integrals is a very cumbersome task, due to the fact that the variable $z$ appears in the argument of the Chebyshev polynomials. Hence, the individual integrals $I_{\alpha,k}(h,\tau)$ and $\hat{I}_{\alpha,k}(h,\tau)$ for $\alpha = 1,\ldots,4$ depend on both variables $h$ and $\tau = ht$. Nevertheless, as it is clear from Fig. 1, the final result $I_k$ in (93) depends only on the scaling variable $\tau = ht$. To show this analytically, one has to use the explicit form of the Chebyshev polynomials and expand the powers of $z$, which then lead to integrals that can be evaluated via [60]

$$\int_{-1}^{1} \frac{d\tilde{\epsilon}}{\pi} (1-\tilde{\epsilon}^2)^{m-1/2}(-\tilde{\epsilon})^n \exp(i\tau\tilde{\epsilon}) = (2m-1)!! \left(i\frac{\partial}{\partial\tau}\right)^n \frac{J_m(\tau)}{\tau^m}, \qquad (96)$$

for arbitrary integers $m$ and $n$. In turn, each of the integrals $I_{\alpha,k}(h,\tau)$ and $\hat{I}_{\alpha,k}(h,\tau)$ can be rewritten as a double sum of terms containing various powers of $h$ and expressions of the form (96). Due to the huge amount of terms appearing, we were able to verify the relation $\frac{\partial}{\partial h} I_k(h,\tau) = 0$ only using Mathematica, for $k < 20$. Using this property, one can also obtain the final result by setting $h = 0$ with $\tau = ht$ fixed in all of the integrals. Then the argument of the Chebyshev polynomials simplifies to $-\tilde{\epsilon}$, the integrals with the hat symbols are identical to the ones without, and both $I_{3,k}$ and $I_{4,k}$ in (95) vanish explicitly. The remaining terms can be evaluated via the integral identities [60]

$$\int_{-1}^{1} \frac{d\tilde{\epsilon}}{\pi} T_{2l}(\tilde{\epsilon}) \frac{\cos(\tilde{\epsilon}ht)}{\sqrt{1-\tilde{\epsilon}^2}} = (-1)^l J_{2l}(ht), \qquad \int_{-1}^{1} \frac{d\tilde{\epsilon}}{\pi} T_{2l+1}(\tilde{\epsilon}) \frac{\sin(\tilde{\epsilon}ht)}{\sqrt{1-\tilde{\epsilon}^2}} = (-1)^l J_{2l+1}(ht), \quad (97)$$

leading to the final result $I_k = J_k^2(ht)$.

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
