# Peer review of "Universal front propagation in the quantum Ising chain with domain-wall initial states"

_SciPost Physics, doi:SciPost Phys. 1, 014 (2016)_

## Round 2 · Referee Report · Anonymous · 2016-11-18

Strengths

1) Interesting results
2) Clearly written

Weaknesses

Some parts of the paper need additional details

Report

This paper studies the melting of domain wall initial states in the ordered phase of the transverse field Ising chain. The two initial states considered, $|{\rm JW}\rangle$ and $|{\rm DW}\rangle$, are domain walls for the order parameter magnetization. They differ in the way they are created starting from one of the symmetry-broken ground states of the model: $|{\rm JW}\rangle$ is created by acting with an operator that is local in terms of the Jordan-Wigner (JW) fermions, and $|{\rm DW}\rangle$ is created by acting with an operator that is non local in terms of the JW fermions. The focus is on the profiles of longitudinal magnetization, transverse magnetization and entanglement entropy (EE). The authors analytically show that the profiles of the longitudinal magnetization generated by evolving the state $|{\rm JW}\rangle$ are, after a rescaling, the same as those found in Ref. [8]. Where the density of particles in a free fermionic chain (XX model) evolving from a state with step occupation was considered. This demonstrates what has been numerically observed in Ref.[29]. The other observables, on the other hand, show different behaviours with respect to the XX model; in particular the EE always saturates for large times as the system is not critical. Starting from $|{\rm DW}\rangle$ instead, the equivalence with the free fermion result does not hold even for the longitudinal magnetization. Some of this results are generalised to the XY model without external field.

I think the paper is very interesting and well written, presenting the results in a clear and easy-to-read manner. In some cases, however, the authors could give a bit more details without compromising the readability of the paper. This would make the paper even more accessible to the inexperienced readers.

Requested changes

Here is a list of questions/corrections I would like the authors to address:

1. In the "Model and Setting" section the authors do not write the explicit form of the coefficients $\psi_k(m)$ and $\phi_k(m)$ and of the energies $\epsilon_k$ (the energy is written only at page 6, also, with a slightly different notation (see point 2)). I think that writing these quantities explicitly can be useful for the reader, for example to understand the semi-classical interpretation of Section III-B. On the same lines, I think some more explanation of the nature of ${|{1}\rangle=\eta_1^\dagger |{0}\rangle}$ could be helpful to understand the physics of the model. Maybe they can anticipate and enrich the discussion after Eq. (15), moving it to the point where they first talk about this state.

2. Some clarification is needed in the initial part of Section III-B, when discussing the semi-classical approach. Strictly speaking the operators $\{\eta^\dagger_q\}$ create eigenmodes of the open chain, which do not have a well defined velocity: they create quasi-particles both moving with velocity $v_q$ and $-v_q$. As the authors correctly point out in Section III-C, the proper way of proceeding would be to decompose $a_{2 n_0-1}$ in terms of the operators creating the modes of the anti-periodic chain. I think they could move the discussion about the anti-periodic chain directly at the beginning of Section III-B and carry out the semi-classical approach using it.

Note that the notation used in Section III-B is slightly different from the one used before. Before $q$ was an integer between $1$ and $N$ while here is a continuous variable, so strictly speaking the energy can not be indicated as $\epsilon_q$ in both cases. Also, when the wavenumber of the open chain "becomes continuous" it varies $[0,\pi]$, the interval $[-\pi,\pi]$ is the one of the anti-periodic (periodic) chain.

3. Why do the authors use anti-periodic boundary conditions in the "Form-Factor approach" of Section III-C? Clearly the domain wall state does not exist for periodic boundary conditions (one would need at least two domain walls), but what about the open boundary conditions considered before? Is it technically more complicated? Is the form factor not known?

4. In Equations (37) and (38) ${\sin (2(n-n_0)+1)\frac{p-q}{2}}$ is really confusing without additional brackets.

5. Can a semi-classical approach also be used in the case of the $|\rm DW\rangle$ state of Section IV? In this case the quasi-particles are emitted in the whole left half-chain and not only at the point $n_0$, but I would still expect a semi-classical reasoning to be applicable.

6. I find some parts of Section V confusing. First, I think that Eq. (48) is defined only for $|v|<1$ (for $|v|>1$ it is meaningless). For $|v|>1$ (i.e. $|n-n_0|>v_{\rm max} t$) the entropy must remain constant. Second, I don't fully understand the claim about the unitary equivalence of the current carrying steady-state and the ground state. Do the authors conjecture that the current carrying steady state is related to the ground state by a unitary transformation which is "local in space"? Since the EE between the two half chains in the late time state is equal to the one in the ground state, I would conclude that the density matrices of the late time state and of the ground state reduced on a half chain could be unitary related. How can I pass from this to a claim on the current carrying stationary state? I think the authors should give more details on this.

7. Section VI is very schematic. I suggest to write the explicit definition of $|\rm JW\rangle_{XY}$ (which is now created by $a_{2 n_0 -3}$).

  • validity: top
  • significance: high
  • originality: high
  • clarity: top
  • formatting: perfect
  • grammar: perfect

Author:  Viktor Eisler  on 2016-12-09  [id 82]

(in reply to Report 1 on 2016-11-18)

We thank the referee for the positive evaluation of our work and for his/her insightful questions and comments. We are happy to see that the referee has found the paper interesting and well-written. Regarding the additional details requested, please find below our point-to-point answers.

1)

In the "Model and Setting" section the authors do not write the explicit form of the coefficients $\psi_k(m)$ and $\phi_k(m)$ and of the energies $\epsilon_k$ (the energy is written only at page 6, also, with a slightly different notation (see point 2)). I think that writing these quantities explicitly can be useful for the reader, for example to understand the semi-classical interpretation of Section III-B.

In fact, we have calculated the eigenvalues and vectors of Eq. (5-6) numerically, which is now explicitly stated in the following text. We are aware of the fact, that for the TI chain the analytical form is also known (i.e. standing wave solutions with proper quantization of the momenta). However, since in our discussion of the open chain we never really use the analytical forms of these quantities, we have decided to omit them.

On the same lines, I think some more explanation of the nature of $|1\rangle=\eta_1^\dagger|0\rangle$ could be helpful to understand the physics of the model. Maybe they can anticipate and enrich the discussion after Eq. (15), moving it to the point where they first talk about this state.

The corresponding part of the manuscript has been rewritten and the discussion of the edge mode has been moved to directly after Eq. (12).

2)

Some clarification is needed in the initial part of Section III-B, when discussing the semi-classical approach. Strictly speaking the operators $\eta^\dagger_q$ create eigenmodes of the open chain, which do not have a well defined velocity: they create quasi-particles both moving with velocity $v_q$ and $−v_q$. As the authors correctly point out in Section III-C, the proper way of proceeding would be to decompose $a_{2n0−1}$ in terms of the operators creating the modes of the anti-periodic chain. I think they could move the discussion about the anti-periodic chain directly at the beginning of Section III-B and carry out the semi-classical approach using it.

For simplicity of the semi-classical argument, in Section III-B we deal with an infinite chain from the beginning on. That is, there are no boundary conditions imposed, and the eigenmodes are just traveling waves with continuous momenta. We have now rewritten the first paragraph of Sec. III-B to stress this issue, which was not made explicit enough in the previous version.

Note that the notation used in Section III-B is slightly different from the one used before. Before q was an integer between 1 and N while here is a continuous variable, so strictly speaking the energy can not be indicated as $\epsilon_q$ in both cases. Also, when the wavenumber of the open chain "becomes continuous" it varies [0,$\pi$], the interval [$−\pi$,$\pi$] is the one of the anti-periodic (periodic) chain.

We use the index $k=1,\dots,N$, whenever we deal with the discrete modes of the open chain. When working in the infinite chain limit, we use $q\in\left[-\pi,\pi\right]$ which is indeed continuous. As stressed above, this is not taken as the thermodynamic limit of the open chain.

3)

Why do the authors use anti-periodic boundary conditions in the "Form-Factor approach" of Section III-C? Clearly the domain wall state does not exist for periodic boundary conditions (one would need at least two domain walls), but what about the open boundary conditions considered before? Is it technically more complicated? Is the form factor not known?

To the best of our knowledge, the form factors are known only for periodic and anti-periodic boundary conditions.

4) Syntax of Eqs. (37) and (38) has been corrected.

5)

Can a semi-classical approach also be used in the case of the |DW⟩ state of Section IV? In this case the quasi-particles are emitted in the whole left half-chain and not only at the point $n_0$, but I would still expect a semi-classical reasoning to be applicable.

Unfortunately, the situation is more complicated. For the semi-classical reasoning to work, we need independent quasi-particles in Fourier-space. This is valid for the JW state, where the single Majorana mode is indeed a superposition of Fourier modes. For the DW, however, one has a factorized form only in real space, see Eq. (10), with Majoranas acting on the left half-chain. It is unclear to us how to reinterpret this as independent quasi-particles in Fourier space.

6)

I find some parts of Section V confusing. First, I think that Eq. (48) is defined only for $|v|<1$ (for $|v|>1$ it is meaningless). For $|v|>1$ (i.e. $|n−n_0|>v_{max} t$) the entropy must remain constant.

That's correct, we have updated the text below Eq. (48) to clarify this issue.

Second, I don't fully understand the claim about the unitary equivalence of the current carrying steady-state and the ground state. Do the authors conjecture that the current carrying steady state is related to the ground state by a unitary transformation which is "local in space"? Since the EE between the two half chains in the late time state is equal to the one in the ground state, I would conclude that the density matrices of the late time state and of the ground state reduced on a half chain could be unitary related. How can I pass from this to a claim on the current carrying stationary state? I think the authors should give more details on this.

We thank the referee for pointing out this issue, which was not properly formulated in the previous version. We have now rewritten the corresponding paragraph and extended the discussion about the steady state. First, it should be pointed out that, inferring from the scaling behaviour of the entropy profiles, the equality with the ground-state entropy for $t\to\infty$ does not only apply to the half chain, but to arbitrary cuts in a finite distance from the centre. This signals that a translational invariant steady state is formed in the bulk. Now, our proper statement is as follows: if we reduce the time-evolved state to this bulk steady-state regime and send $t \to \infty$, the result is unitarily equivalent to the reduced density matrix of the ground state. We have added Eq. (49) with a precise mathematical formulation of this conjecture.

7) We have now defined $|JW\rangle_{XY}$ explicitly, see Eq. (58).

---

## Round 2 · Referee Report · Anonymous · 2016-11-21

Strengths

1- very interesting problem
2- clear presentation
3- self-consistent work

Weaknesses

1- some interpretations are questionable/unclear
2- a point that I find very important is hidden between the lines

Report

The authors consider the time evolution of two inhomogeneous states, JW and DW, under the Hamiltonian of the quantum Ising chain in the ferromagnetic phase. The initial state is obtained acting with spin flip with respect to z on one half of the (symmetry-breaking) ground state; in addition, in the JW case, the spin at the middle is flipped with respect to x. Observables display a nontrivial profile as a function of the ratio position over time. The authors prove that, in the JW case, the magnetisation profile is, to some extent, universal: after a simple rescaling by the maximal velocity, it does not depend on the magnetic field and is the same as in the free fermion case. On the other hand, the time evolution of DW does not exhibit similar universal properties. The authors provide both a semiclassical explanation and an exact derivation of the results. Finally, they consider the time evolution of the entanglement entropy of a bipartition.
I think that the problem considered is very interesting and the presentation is clear, so I strongly recommend this paper for publication in SciPost.
Apart from minor comments, I have two main observations.
The authors show that the time evolution of two states which are identical in the bulks is characterised by different profiles of observables. That is to say, the expectation values are different in the scaling limit of infinite time, infinite position, and fixed ratio of them. This is very unexpected, indeed it is generally believed that, in the aforementioned limit, the only relevant information about the initial state are its bulk properties. However, as far as I can see, JW and DW are locally indistinguishable in the bulks. I really think that the authors should stress this point much more than they already did.
The second observation is related to their interpretation of the entanglement entropy saturating at long times. The authors are not surprised by such behaviour because the model is not critical. I do not think that this is sufficient to that conclusion. For example, considering the time evolution of a state obtained joining together the ground state and the state with maximal energy (for the sake of simplicity, let’s assume that we are in the paramagnetic phase), I do expect a (locally) critical infinite-time limit.

Requested changes

1- I suggest to summarise the properties of the excitation $\eta_1^{\dagger}$ just after (12).
2- I suggest to use a different notation for the vectors (e.g. $\vec H$ instead of $H$).
3- Since JW is not an excited state, I am not sure that the term "JW excitation" is appropriate.
4- Apparently, Section 3.B is using different notations (e.g., $q$ is now a real variable).
5- There is a typo in (37) and (38): the parentheses of the $\sin$ should be closed outside $\frac{q-p}{2}$.
6- What does it mean that one state is unitarily equivalent to another? Any pure state is. Do the authors mean that the two states are locally related to one another? Can this be inferred just from the behaviour of the entanglement entropy of bipartitions?
7- The authors propose that the infinite-time limit could be locally related to the symmetry-unbroken ground state. Can the authors comment on the fact that such state does not have cluster decomposition properties? Could the authors check that the two-point function of the order parameter does no vanish in the infinite-time limit?

  • validity: high
  • significance: high
  • originality: high
  • clarity: high
  • formatting: excellent
  • grammar: excellent

Author:  Viktor Eisler  on 2016-12-09  [id 81]

(in reply to Report 2 on 2016-11-21)

We thank the referee for the careful reading of the manuscript and for the insightful comments. We were pleased to see that he/she has found our work very interesting. In our response, we first clarify the two main issues raised by the referee.

Apart from minor comments, I have two main observations. The authors show that the time evolution of two states which are identical in the bulks is characterised by different profiles of observables. That is to say, the expectation values are different in the scaling limit of infinite time, infinite position, and fixed ratio of them. This is very unexpected, indeed it is generally believed that, in the aforementioned limit, the only relevant information about the initial state are its bulk properties. However, as far as I can see, JW and DW are locally indistinguishable in the bulks. I really think that the authors should stress this point much more than they already did.

One should be careful when talking about bulk properties in the present setting. Indeed, in the bulk (that is far away from the domain wall location) both initial states are just symmetry broken ground states of the system, and it is the very presence of the domain wall that induces time evolution. Hence, it is not entirely surprising that the time-evolved profiles will be sensitive to the details of how this domain wall is realized. In fact, the two initial states cannot be considered identical on the level of observables, as JW flips the $\sigma^z$ in the center of the chain, while DW does not. It should be stressed that, since the profiles have a non-analytic behaviour in the initial state, this should not be considered as a bulk property.

The second observation is related to their interpretation of the entanglement entropy saturating at long times. The authors are not surprised by such behaviour because the model is not critical. I do not think that this is sufficient to that conclusion. For example, considering the time evolution of a state obtained joining together the ground state and the state with maximal energy (for the sake of simplicity, let’s assume that we are in the paramagnetic phase), I do expect a (locally) critical infinite-time limit.

We believe that the signatures of the underlying Hamiltonian should be traceable in the entanglement evolution if one considers initial states that have low energies. Then it is only the low-lying part of the spectrum that has significant contribution to the time evolution, which should be sensitive to the presence or absence of a gap in the spectrum. Considering initial states with high energies, the full spectrum will contribute, and thus the criticality of the Hamiltonian (which is a low-energy property) should not play a special role. In fact, when considering global quenches of the Ising chain, it has been observed that the entanglement growth is always linear, independently of whether the final Hamiltonian is critical or not.

We now proceed to address the detailed comments of the referee, indicating corresponding changes to the manuscript.

1)

I suggest to summarise the properties of the excitation $\eta_1^\dagger$ just after (12).

According to the referee's suggestion, we have reorganized the text and the properties of the edge mode are now discussed right after Eq. (12).

2)

I suggest to use a different notation for the vectors (e.g.$\vec{H}$ instead of $H$)

The overline vector notation would not be very convenient, as in some parts of the manuscript we use $\tilde H$ symbols as well, which would then look awkward with the additional overline. Thus we prefer to keep the original notation.

3)

Since JW is not an excited state, I am not sure that the term "JW excitation" is appropriate.

We are not sure we understand this comment. In our terminology, everything that is above the ground state is an excitation.

4)

Apparently, Section 3.B is using different notations (e.g., q is now a real variable).

We use the letter $k$, whenever we deal with the discrete modes of the open chain. When working in the thermodynamic limit we use $q$, which is indeed a continuous momentum variable. We have reformulated the initial part of Sec. 3.B to make it clear, that we are explicitly working in the infinite chain limit there.

5) Syntax of Eqs. (37) and (38) has been corrected.

6)

What does it mean that one state is unitarily equivalent to another? Any pure state is. Do the authors mean that the two states are locally related to one another? Can this be inferred just from the behaviour of the entanglement entropy of bipartitions?

We thank the referee for this question. In fact, our formulation of this statement was somewhat sloppy in the previous version. We have now rewritten the complete paragraph where the steady state is discussed and the statement has been made precise. In fact, we conjecture that if we reduce the time-evolved state to the translationally invariant steady-state regime in the bulk of the chain and send $t \to \infty$, the result is unitarily equivalent to the reduced density matrix of the ground state. We have added Eq. (49), where this statement is precisely formulated.

7)

The authors propose that the infinite-time limit could be locally related to the symmetry-unbroken ground state. Can the authors comment on the fact that such state does not have cluster decomposition properties? Could the authors check that the two-point function of the order parameter does no vanish in the infinite-time limit?

This is indeed a very interesting question. In fact, the Pfaffian approach can easily be generalized for the calculation of two-point functions and there is some hope to extend the form-factor based calculations as well. As we already pointed out at the end of our discussion in Sec. VII, it is our primary goal to address such questions in our future research of this topic.

---

## Round 3 · Author Response

See response letters to referees

---

## Editorial Decision

published